# Curvature-processing domains in primate V4

**Rendong Tang[†], Qianling Song[†], Ying Li, Rui Zhang, Xingya Cai, Haidong D Lu\***

State Key Laboratory of Cognitive Neuroscience and Learning, IDG/MGovern Institute for Brain Research, Beijing Normal University, Beijing, China

**Abstract** Neurons in primate V4 exhibit various types of selectivity for contour shapes, including curves, angles, and simple shapes. How are these neurons organized in V4 remains unclear. Using intrinsic signal optical imaging and two-photon calcium imaging, we observed submillimeter functional domains in V4 that contained neurons preferring curved contours over rectilinear ones. These curvature domains had similar sizes and response amplitudes as orientation domains but tended to separate from these regions. Within the curvature domains, neurons that preferred circles or curve orientations clustered further into finer scale subdomains. Nevertheless, individual neurons also had a wide range of contour selectivity, and neighboring neurons exhibited a substantial diversity in shape tuning besides their common shape preferences. In strong contrast to V4, V1 and V2 did not have such contour-shape-related domains. These findings highlight the importance and complexity of curvature processing in visual object recognition and the key functional role of V4 in this process.

**\*For correspondence:**
haidong@bnu.edu.cn

[†]These authors contributed equally to this work

**Competing interests:** The authors declare that no competing interests exist.

## Introduction

Shape extraction is crucial for object recognition and is a major function of the primate visual cortex. This process has been reported to occur in the ventral visual pathway (V1-V2-V4-IT) in a hierarchical manner (*Connor et al., 2007*). The mid-level area V4 plays an important role in this process (*Roe et al., 2012*; *Pasupathy et al., 2020*). Many V4 neurons selectively respond to curvatures (*Pasupathy and Connor, 1999*) and complex shape features (*Desimone and Schein, 1987*; *Gallant et al., 1993*; *Kobatake and Tanaka, 1994*). However, the distribution of these shape-selective neurons in V4 remains unclear.

In the V1-V2-V4-IT pathway, orientation maps have been reported in V1, V2, and V4. Functional maps for faces (*Wang et al., 1996*; *Tsao et al., 2006*) and various shape features (*Fujita et al., 1992*; *Tsunoda et al., 2001*; *Sato et al., 2009*) have been reported in IT. This processing hierarchy appears to lack functional structures for shape features of intermediate complexity, including curvatures and angles. A previous fMRI study reported several patches of the ventral cortex that prefer curved features with one being located in V4 (*Yue et al., 2014*). However, given the limited spatial resolution of fMRI signals, these patches were relatively large (at the cm-scale). It remains unclear whether there are submillimeter functional columns for curvatures that are similar to those for edge orientations.

## Results

### Curvature domains imaged using intrinsic signal optical imaging (ISOI)

Through ISOI on anaesthetized macaques, we imaged cortical hemodynamic signals for various contour features. The stimulus set included simple contour shapes, including circles and triangles, as well as parts of these shapes, including curves, angles, and short straight lines. As shown in *Figure 1C*, the contours were presented on a full screen and drifted along one of four or eight

**Figure 1.** Intrinsic signal optical imaging (ISOI) of curvature domains in area V4. (A) A schematic representation of the macaque brain showing the imaging region (green circle) and sulci locations. lu, lunate sulcus; st, superior temporal sulcus; and io, inferior occipital sulcus. (B) In vivo image of the blood vessel pattern in the 16 mm diameter imaging region in Case 1, which included parts of V1, V2, and V4. The exposed V4 region was between the st and lu sulci. A, anterior; M, medial. (C) An illustration of the full-screen stimulus pattern used for ISOI imaging. Circle diameter: 2.5°. Drifting speed: 4°/s. (D–L) Functional maps (SVM maps) from Case 1. The icons shown at the top represent the stimulus conditions being compared. In the ISOI maps, dark and white regions were preferentially activated by the stimulus icons shown on the left and right, respectively. For the maps shown in G, H, J, and K, data from different stimulus orientations were pooled for lines, curves, and angles. Other maps were obtained with the exact stimuli as shown on the maps. (D) The circle vs. triangle map shows clear patches in V4 (dark regions preferred circles and white regions preferred triangles), which were absent in V1 and V2. The dotted line represents the border between V1 and V2. (E) The orientation preference map shows the 45° (dark) and 135° (white) orientation domains in V2 and V4. The lack of orientation domains in V1 could have been resulted from the low SF of the stimulus gratings (0.25 c/ degrees). (F) The color preference map shows the color domains in V1, V2, and V4. (G) The circle vs. straight line map shows a preference pattern similar to that in the circle vs. triangle map (D). (H) The curve vs. straight line map shows patterns similar to those in D and G. The curve was a half-circle. (I) The curve-orientation map shows smaller and weaker patches preferring different curve orientations. (J) The triangle vs. straight line map shows weaker and larger patches than those in the circle vs. straight line map in G. The dark domains preferred triangles over lines and occupied the same regions as the white patches in D. (K) The angle vs. straight line map shows a very weak pattern similar to that in J. (L) There was no clear pattern in the angle-orientation map.

The online version of this article includes the following figure supplement(s) for figure 1:

**Figure supplement 1.** Comparison of single-condition maps and difference maps (Case 1).

**Figure supplement 2.** Additional functional maps of Case 1.

directions. Each stimulus condition was presented for 3.5 s and repeated 25–50 times in a random sequence. The light reflectance was imaged through an optical chamber covering parts of V1, V2, and V4 (*Figure 1A and B*). The exposed V4 region corresponded to the visual field of 0°−10° eccentricity in a lower quadrant. We obtained support vector machine (SVM) maps that compared cortical images collected during two stimulus conditions (*Figure 1D–L*). In the circle vs. triangle map (*Figure 1D*), the black and white patches corresponded to regions that preferred circles and triangles, respectively. This V4 pattern significantly differed from the orientation (*Figure 1E*) and color patterns (*Figure 1F*) obtained with established methods (*Tanigawa et al., 2010*; *Li et al., 2013*). In contrast to V4, we did not observe circle vs. triangle patterns in V1 and V2, or in the cortex anterior to the superior temporal sulcus (*Figure 1D*). We also examined single-condition maps (stimulus vs.

blank); however, the responses were dominated by feature-non-specific activation and the shape-selective response patterns were not apparent (*Figure 1—figure supplement 1*).

Similar V4 patterns were observed in the circle vs. triangle (*Figure 1D*), circle vs. line (*Figure 1G*), and curve vs. line maps (*Figure 1H*). Although the strength of the dark patches in the latter two maps appeared weaker, they had similar general layouts and locations as the first one (also see *Figure 1—figure supplement 2O–R* for enlarged views). This similarity suggests that the key contour feature responsible for these dark patches was the curviness. Therefore, we have henceforth referred to these patches as 'curvature domains'. These curvature domains contained smaller substructures. For example, a map comparing differently oriented curves showed smaller and weaker patches within the curvature domains (*Figure 1I*, *Figure 1—figure supplement 2*). Note that this curve orientation (also called direction) is different from the grating orientation (0–180˚) and has a range of 0–360˚.

Compared with the curvature domains, triangle-activated cortical regions appeared significantly weaker. In the triangle vs. line (*Figure 1J*) and angle vs. line maps (*Figure 1K*), the dark patches (angle-preferring) were apparently more diffused and weaker than those in the circle vs. line or curve vs. line maps (*Figure 1G and H*). This was also observed in the circle vs. triangle map (*Figure 1D*), which had fewer and weaker white patches (triangle-preferring) than dark patches (circle-preferring) (also see *Figure 1—figure supplement 2G*). Moreover, the orientation subdomains observed in the curve-orientation map (*Figure 1I*) were not present in the angle-orientation map (*Figure 1L*).

Imaging results of seven hemispheres (cases) of six macaque monkeys were consistent (*Figure 2B*, *Figure 2—figure supplement 1B*). We observed curvature domains in the entire exposed V4 surface representing both foveal and peripheral visual fields, but peripheral V4 tend to prefer larger circles (*Figure 2—figure supplement 2A–D*). Moreover, similar patterns can be repeatedly imaged from the same chamber over months (*Figure 2—figure supplement 3*). Modification of the experimental conditions (e.g. monocular left- or right-eye viewing conditions or binocular conditions, see *Figure 1—figure supplement 2M and N*) and noncritical stimulus parameters (e.g. contour brightness, filled disks, and regular vs. random contour arrangements) did not alter the main pattern features (*Figure 2—figure supplement 2E–I*).

The curvature domains had an average diameter of 518 ± 25 μm (mean ± s.e.m), which was similar to that of the orientation domains (479 ± 16 μm, p=0.074, paired t-test, n = 7; *Figure 2E*). The curvature domains occupied 12.1 ± 1.7% of the V4 surface, smaller than the coverage of the orientation domains (22.3 ± 3.7%, p=0.025, paired t-test, n = 7). The intrinsic optical signal amplitude (percentage change), which was measured from the raw subtraction maps, was numerically, but not significantly, larger for the circle vs. triangle maps than the orientation maps (curvature domains: 0.036%, orientation domains: 0.029%, p=0.15, paired t-test, n = 7, *Figure 2F*).

Both the curvature and orientation domains represent shape features; therefore, it is logical to expect that they might overlap. However, we observed little overlap between the curvature and orientation domains (*Figure 2D*). The overlap size was significantly smaller than that expected in a random distribution (p=0.03, paired t-test, n = 7, *Figure 2G*), which indicates an avoidance tendency between both domains. Moreover, the curvature domains did not overlap with the color domains (p=0.004, paired t-test, n = 7; *Figure 2—figure supplement 1F*). Contrastingly, the triangle-preferring white patches in the circle vs. triangle maps tended to overlap with the orientation (p=0.003, paired t-test, n = 7; *Figure 2—figure supplement 1F*) but not the color domains (p=0.17, paired t-test, n = 7; *Figure 2—figure supplement 1F*).

## Two-photon calcium imaging of the responses to contour shapes

ISOI used a metabolism-based hemodynamic signal; therefore, the actual neuronal responses and neuron constitution within these domains were unclear. Moreover, ISOI could not provide information regarding the variations in different cortical layers. We then used two-photon calcium imaging to address these questions. Based on the ISOI functional maps, we injected AAV1/9-GCaMP6s virus into multiple locations in two chambers (Cases 3 and 4 in *Figure 2A*, *Figure 3—figure supplement 1*). The injections targeted either the centers of the curvature domains or regions outside these domains. We imaged the monkeys under similar anesthesia conditions 1.5 months after the injections. *Figure 3C* shows an image from Case 3 (red frame in *Figure 3A and B*, also see *Figure 3—video 1*) at a 210 μm depth. The center of this image frame contained a curvature domain (*Figure 3B*). Mapped with both manual and computer-controlled visual stimuli, the population

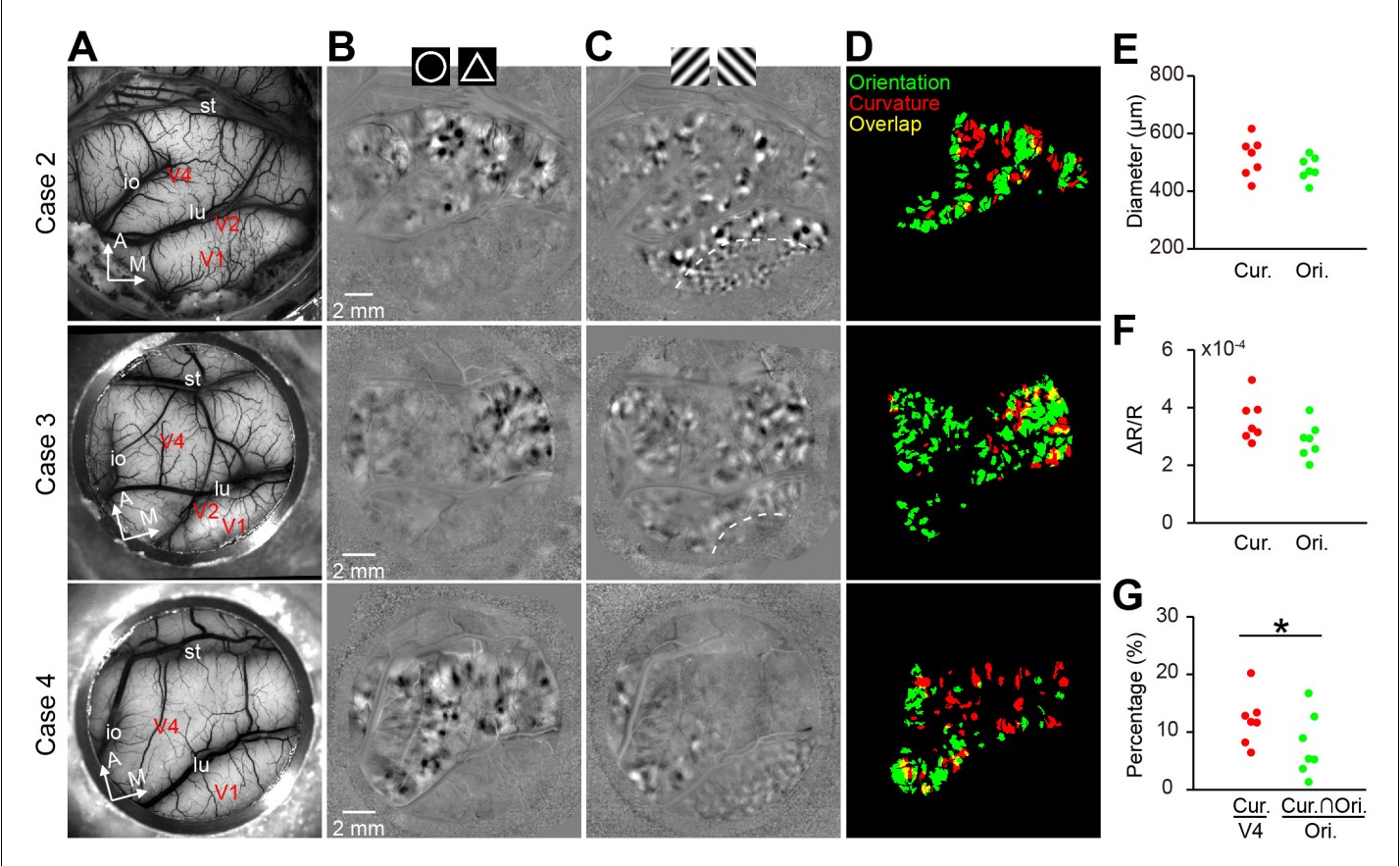

**Figure 2.** Comparison of curvature and orientation maps. (**A–D**) Maps from Cases 2–4 are shown in rows 1–3. (**A**) The blood vessel patterns of the imaging regions. (**B**) The circle vs. triangle maps obtained using the same stimuli as those for *Figure 1D*. (**C**) Orientation preference maps showing the 45˚ vs. 135˚ orientation patterns in V1, V2, and V4. The dotted lines represent the borders between V1 and V2. Case 4 did not have V2 exposed on the surface. (**D**) The spatial relationship between the curvature and orientation domains (0˚, 45˚, 90˚, 135˚) in each case. (**E**) The curvature and orientation domains had similar domain sizes. (**F**) The curvature and orientation domains had similar response amplitudes. (**G**) The size of the overlap regions between the curvature and orientation domains was smaller than that of the random prediction (p=0.030; paired t-test, n = 7), which indicates a tendency of separation of these two types of domains.

The online version of this article includes the following figure supplement(s) for figure 2:

**Figure supplement 1.** Three types of maps from all seven cases.

**Figure supplement 2.** Circle vs. triangle maps obtained with different stimulus parameters.

**Figure supplement 3.** Repeated imaging of the maps on different days.

receptive field (RF) of this region was ~4˚ in size and approximately 7˚ from the fovea (*Figure 4—figure supplement 1A*).

We tested 19 different curved and rectilinear contours, presented at different orientations, which resulted 73 unique stimuli (Figure 6A, also see Materials and methods). Each stimulus contained a single contour element similar to those used in the ISOI experiments (*Figure 3—video 1*). The stimulus moved across the population RF along a direction randomly chosen. Consistent with the ISOI results (*Figure 3B*), we observed significant activation of the neurons by circle stimuli (*Figure 3D*). The whole frame showed a scaled-down response to triangles (*Figure 3E*). The results demonstrate that different stimuli types mainly affected the population response amplitudes rather than activating different cells within the curvature domain.

We determined the preferences of curved and rectilinear stimuli for each pixel in the two-photon image (*Figure 4B*). This imaged region showed a preference for curved (red) over rectilinear (green) stimuli. Consistent preference patterns were observed for this two-photon map (*Figure 4B*) and the ISOI map from the same region (*Figure 4A*). Similar preference patterns were observed at deeper

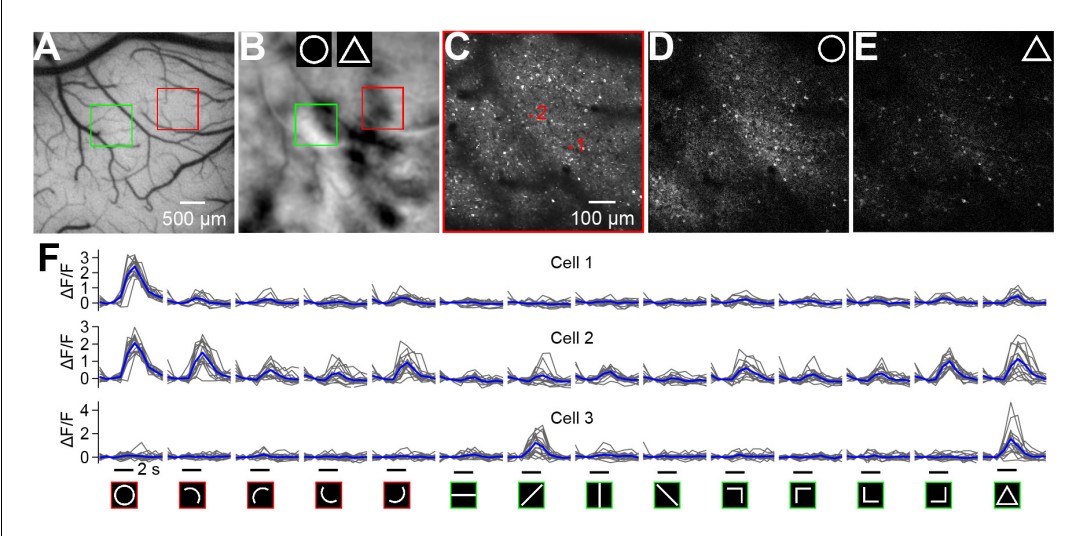

**Figure 3.** Two-photon calcium imaging of neuronal shape responses. (**A**) An image of the cortical region obtained from Case 3 in which AAV-GCaMP6s was injected for calcium imaging. The red and green frames indicate two regions examined using two-photon imaging. (**B**) Circle vs. triangle map of the region shown in A. (**C**) Neurons in the red-frame region shown in A and B that were imaged using a 16× objective at a 210 μm depth from the cortical surface. Two neurons marked in red are examined in detail in panel F. Scale bar applies to C–E. (**D and E**) Single-condition response maps (ΔF) for the circle (**D**), and triangle (**E**) stimuli. Each map was obtained after averaging 15 repeats and pooling of 4 orientations. (**F**) Responses (ΔF/F) of 3 neurons to 14 typical contour stimuli (showed at the bottom, red: curvature stimuli; green: rectilinear stimuli). The responses to circle and triangle were the best-orientation ones. Cells 1 and 2 were selected from the curvature domain shown in C. Although both neurons preferred circle stimuli, they exhibited different response levels to other stimuli types. Cell 3 was selected from a rectilinear region (marked in *Figure 4F*) and was strongly activated by the triangle stimulus and one of its line segment. The gray and blue lines represent individual trials and the average, respectively. The stimulus duration (2 s) is labeled at the bottom of the last row.

The online version of this article includes the following video and figure supplement(s) for figure 3:

**Figure supplement 1.** All the 10 two-photon-imaged V4 regions in this study.

**Figure 3—video 1.** Fluorescent responses of V4 neurons to contour shapes.

https://elifesciences.org/articles/57502#fig3video1

---

layers for this (*Figure 4B and C*) and all five locations imaged at multiple depths (*Figure 4D–I*, *Figure 3—figure supplement 1*). Therefore, the curvature domains form a columnar structure within at least the top V4 layers.

We also performed PCA analysis on the response patterns of all the neurons. This data-driven analysis also revealed a major difference in curved and rectilinear responses, and its association with the neurons' locations that either inside or outside the curvature domains (*Figure 5*).

Although neurons inside the domains exhibited a common preference for curve stimuli, individual neurons showed diverse response properties. *Figure 3F* shows the responses of 3 example neurons to 14 typical stimuli. Two neurons were selected from the curvature domain showed in *Figure 3C* (red markers). Both neurons were strongly activated by circle stimuli; however, cell 2 was also activated by multiple other stimuli. Cell 3 were selected from a region highly activated by rectilinear stimuli (*Figure 4F* arrow) and it showed strong responses to the triangle as well as a 45°-orientated line that resembled an edge of the triangle. Thus, at cellular level, contour-tuning differences exist not only between cells belonging to different functional domains but also between cells within a functional domain. This is further explored in *Figure 6*.

We identified 1923 neurons in 10 two-photon imaged locations (*Figure 3—figure supplement 1*); among them, 788 neurons were in the curvature domains while 1135 neurons were outside of these domains. Despite certain variability due to alignment error, a comparison of the neurons' calcium responses and their corresponding pixel responses in ISOI maps revealed a general consistency between the two measurements (*Figure 4J and K*). *Figure 4J and K* also shows that more neurons had a greater response to curved stimuli than to rectilinear ones (more neurons had a positive Y axis value in both figures). Overall, the neurons with a stronger response to curved stimuli tended to

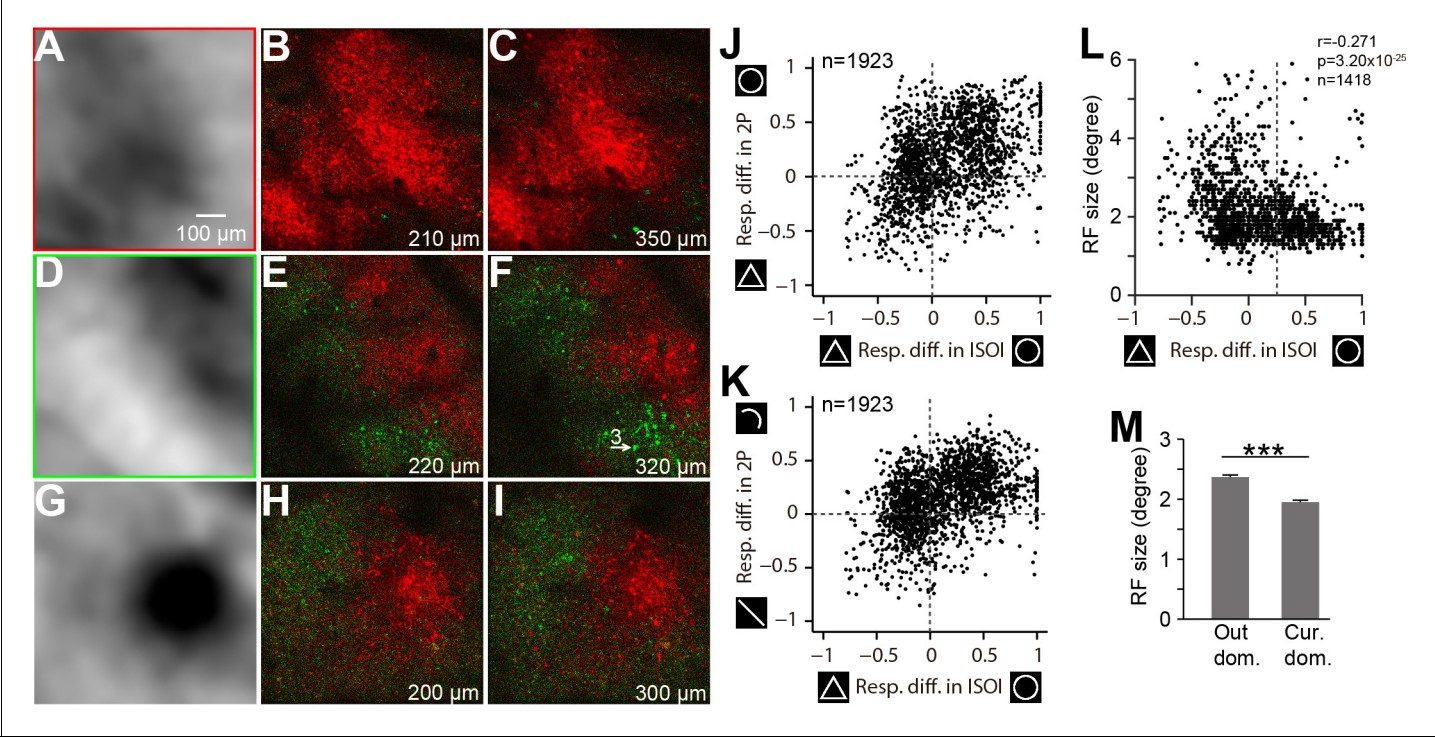

**Figure 4.** Consistency between ISOI and two-photon imaging results. (**A**) An enlarged view of a curvature domain in an ISOI circle-triangle map in Case 3 (the red-framed region in *Figure 3B*). (**B**) A curvature (red) vs. rectilinear (green) preference map for the same image used in *Figure 3C*. Each pixel is color coded for its preferred responses to the curvature (red-framed icons in *Figure 3F*) or rectilinear stimuli (green-framed icons in *Figure 3F*). The brightness of the pixels is proportional to the fluorescence strength and shape preference (see Materials and methods). (**C**) Preference maps for the same location shown in B but obtained from a deeper layer (350 µm from the surface). Contour-type preferences are similar as shallower depths (B). (**D–F**) As in A-C, but for another location in Case 3 (green-framed regions in *Figure 3A and B*). This region contained a subregion preferring curved stimuli (top right) and a subregion preferring rectilinear stimuli (lower left). Two-photon preference maps show consistent contour-type preferences with the ISOI map, as well as consistency between different depths. Cell 3 with an arrow was also used in *Figure 3F*. (**G–I**) As in above two sites, a third example obtained from Case 4, which contained a small curvature domain. Consistency between ISOI imaging and two-photon imaging, as well as consistency between different depths of two-photon images are evident. (**J**) Comparison of the ISOI and two-photon responses. The Y-axis represents differences in the fluorescent responses (optimal orientation) to circles and triangles in all neurons. The X-axis represents differences in hemodynamic responses for corresponding neuron pixels in the ISOI circle vs. triangle maps. There was a significant correlation between the two types of responses (Pearson r = 0.42, p=9.45×10$^{-83}$, n = 1923). (**K**) Similar to J, there was a significant correlation between the fluorescent responses (optimal orientation) to curves vs. lines and ISOI responses to circles vs. triangles (Pearson r = 0.43, p=9.35×10$^{-88}$, n = 1923). (**L**) The RF sizes of all neurons measured (Y-axis) had a negative correlation with the circle-triangle preferential responses of the corresponding pixels in the ISOI imaging (X-axis, same as the X-axis values in J and K). The dotted vertical line represents the threshold chosen (2SD) in determining whether a pixel is inside the curvature domains (the right side of the line) or outside (the left side of the line). (**M**) The average RF size was smaller for neurons inside the curvature domains (1.95 ± 0.032°, n = 583) than those outside (2.37 ± 0.038°, n = 835, p=1.15×10$^{-14}$, t-test). Error bar: s.e.m.

The online version of this article includes the following figure supplement(s) for figure 4:

**Figure supplement 1.** RF sizes of the sampled V4 neurons.

have smaller RFs (*Figure 4L and M*, *Figure 4—figure supplement 1B-E*) and greater surround suppression (*Figure 4—figure supplement 1F and G*), which is consistent with previous findings on the relationship between end-stopping properties and curvature selectivity (*Hubel and Wiesel, 1965*; *Dobbins et al., 1987*; *Ponce et al., 2017*).

## Microarchitectures of the curvature domains

Curvature domains imaged with curves and circles in ISOI were generally similar (e.g. *Figure 1G and H*). However, individual neurons inside the curvature domain often showed different responses to these two stimuli. *Figure 6A,C and E* shows response matrices of three example neurons selected from the curvature domains. Each neuron showed a wide response spectrum to the 73 contour stimuli. To analyze different responses to curves and circles, we labeled neurons that

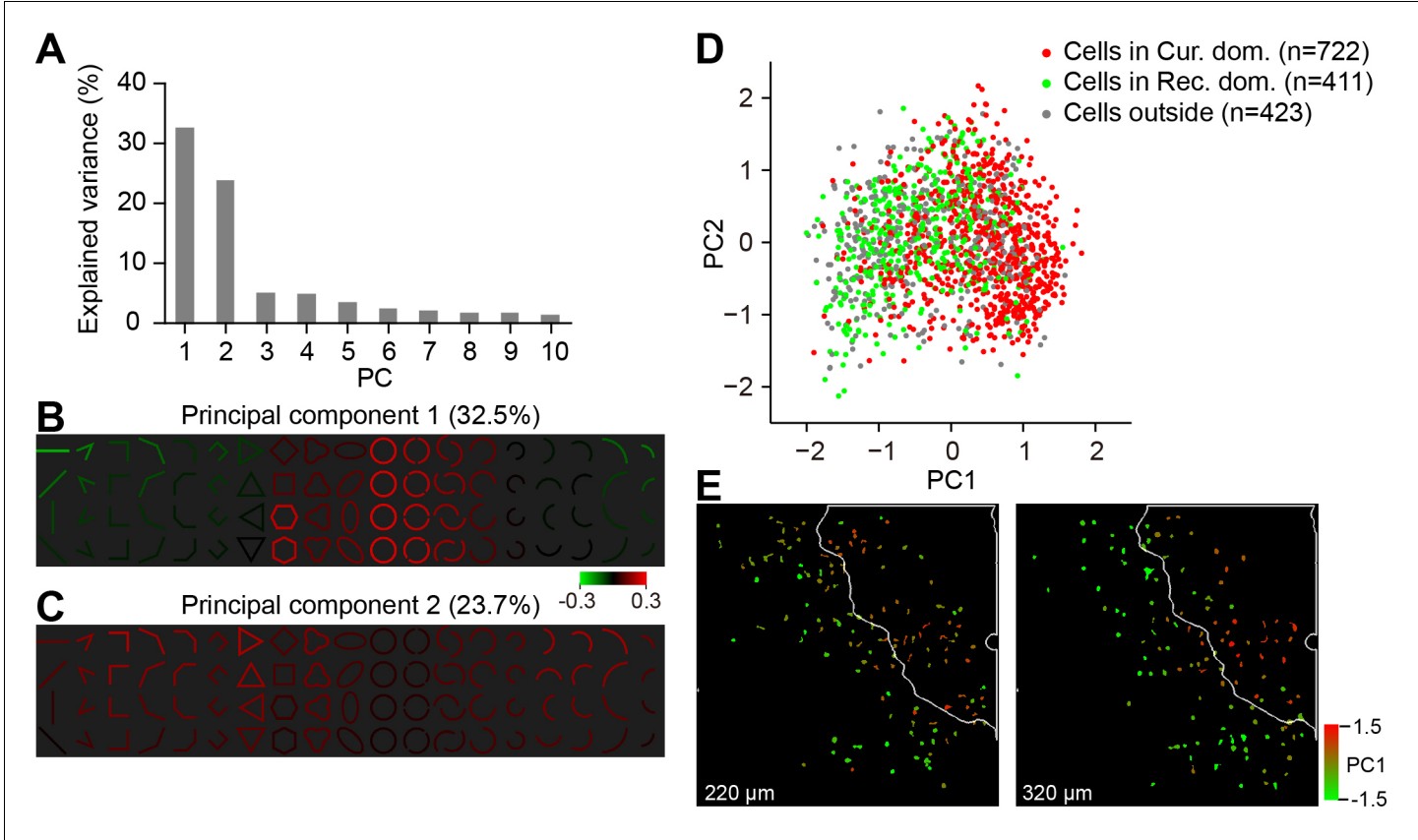

**Figure 5.** PCA results show different response patterns for neurons inside and outside the curvature domains. (**A**) Percentages of the response variance accounted for by the top 10 principle components obtained from the PCA analysis on neurons' response matrices (n = 1556). Stimulus orientations are sorted according to the responses before the PCA analysis. (**B**) PC1 shows a negative response relationship between cirlces and rectilinear lines, comfirming the ISOI findings. The color of each stimulus represents the sign and strength of the population response to that stimulus correlated to this principle component. PC1 explained 32.5% of the response variance. (**C**) As in B, for PC2, which shows positive contribution from many stimuli except for the circles, PC2 explained 23.7% of the variance. (**D**) All neurons (n = 1556) plotted according to their PC1 and PC2 coordinates. Although had a large overlap, curvature neurons (red) tended to separate from the rectilinear neurons (green). Neurons in gray were those located outside of these two types of domains. (**E**) An example two-photon site in which neurons were labeled according to their PC1 coordinates. Neurons with positive PC1 coordinates (red) were mostly found inside the curvature domain on the top right cornor (with white outline). Neurons with negative PC1 coordinates (yellow and green) were mostly located outside the curvature domain.

exhibited significant orientation tuning to curves as 'curve-orientation-preferring neurons' (e.g. *Figure 6A and B*) and those with stronger responses to circles than to curves as 'circle-preferring neurons' (e.g. *Figure 6E and F*). Some neurons that met both criteria were labeled 'dual-preference neurons' (e.g. *Figure 6C and D*). Note that after dual-preference neurons were isolated into a separate group, the earlier two groups no longer contained dual-preference neurons. PCA analysis of response patterns for neurons inside curvature domain also showed differences between circle-preferring neurons and curve-orientation-preferring neurons (*Figure 6—figure supplement 1*). Among the 788 neurons imaged inside the curvature domains, circle-preferring comprised 25.6% (202), dual-preference comprised 14.4% (113), and curve-orientation-preferring neurons comprised 19.2% (151), with the rest not passing the statistical tests. The corresponding proportions of these three groups in the 1135 neurons outside the curvature domains were 7.3% (83), 1.8% (20), and 11.7% (133), respectively (*Figure 6I*). As expected, the summed proportion of these three types of neurons was larger inside the curvature domains (59.1%) than the outside (20.8%). The ratio of circle-preferring to curve-orientation-preferring neurons also appeared to be larger inside (1.35) than outside (0.62) the curvature domains.

When neurons were labeled according to their preferences, there was a tendency of spatial clustering according to their types (*Figure 6G*). Moreover, the curve-orientation-preferring neurons

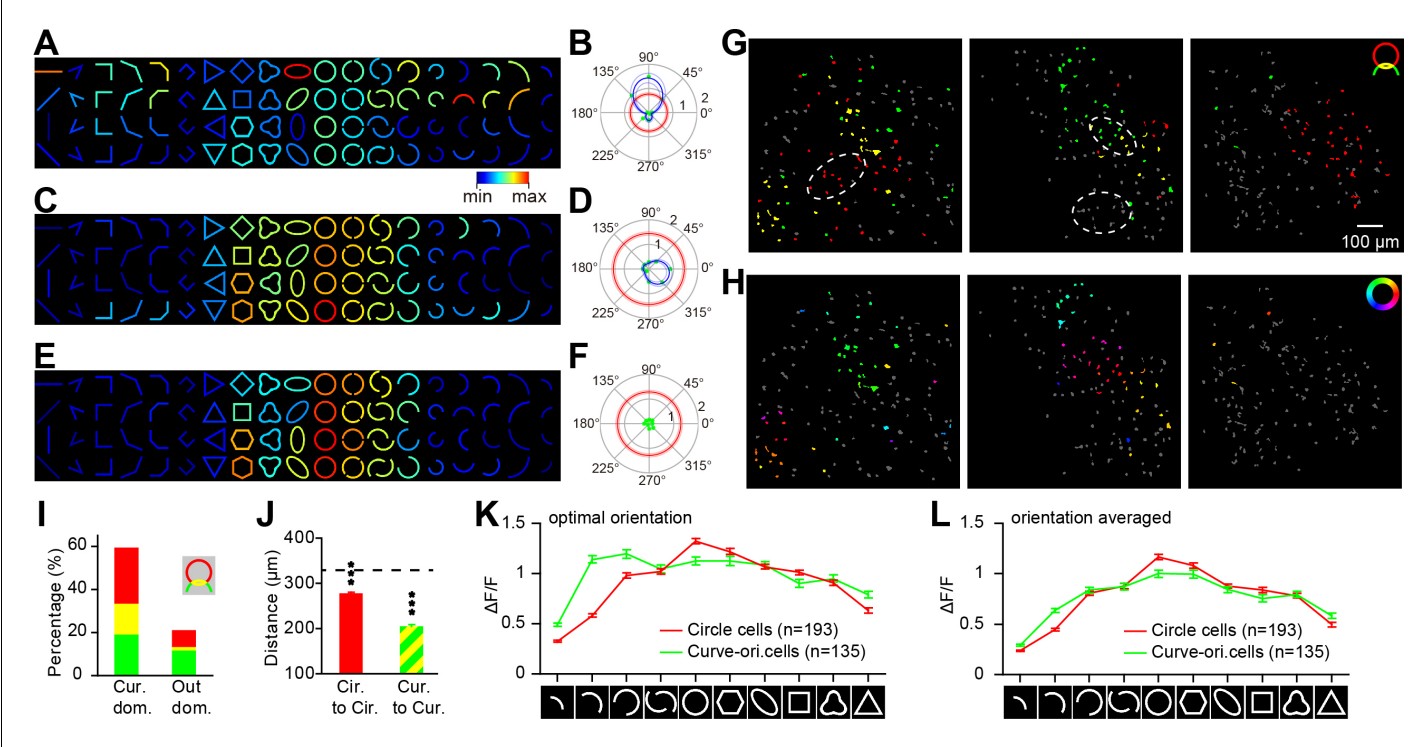

**Figure 6.** Microarchitectures inside the curvature domains. (**A**) Response matrix of an example 'curve-orientation-preferring neuron', which showed strong preference for a half circle oriented at 90˚ and an oval contained a similar curve fragment. (**B**) Orientation tuning curve for curves (half circles). Green dots represent response values to the curves at eight different orientations. Blue line represents fitting curve for the responses. Red circle represents the neurons' average response level (a single value) for the circle stimuli. Pale-blue and red lines represent ± s.e.m of the responses. (**C, D**) As in A and B, but for a 'dual-preference neuron'. This neuron showed significant orientation tuning to curves, along with a maximal response for the circle. (**E, F**) As in A-D, but for a 'circle-preferring neuron'. This neuron responded best to the circles, but did not exhibit curve-orientation tuning. There was no fitting curve for the neuron shown in F due to its weak responses. (**G**) Neurons tended to cluster according to their preferences for circles or curves. Three maps are from three different two-photon imaged locations, and neurons are color coded as shown in the icon: red: circle-preferring neurons; green: curve-orientation-preferring neurons; yellow: dual-preference neurons. Neurons in gray did not pass the circle or curve-orientation preference tests. The neurons in the dotted ovals are further examined in *Figure 7*. (**H**) Neurons showed curve-orientation tuning (including curve-orientation-preferring and dual-preference neurons) tended to cluster according to their preferred orientations. The color of the neurons represents their preferred orientation (0–360˚). (**I**) The percentages of the three neuron types inside and outside the curvature domains. The color code is the same as that in G. (**J**) The average cell-to-cell distances for circle-preferring neurons (red) and neurons showing curve-orientation tuning with similar preferred orientations (differences < 45˚, green and yellow) were shorter than the overall average distance (dotted line) (p<0.001, t-test). Error bar: s.e.m. (**K**) Population averaged fluorescent responses to different contours. For each contour, the response to the optimal orientation was used. The circle-preferring neurons (red) showed gradual increase of response with the length or completeness of the circle, while the curve-orientation-preferring neurons (green) did not. Error bar: s.e.m. (**L**) As in K, but used orientation-averaged responses.

The online version of this article includes the following figure supplement(s) for figure 6:

**Figure supplement 1.** PCA results for neurons inside curvature domains.

**Figure supplement 2.** Spatial periodicity of curve-orientation subdomain.

further clustered based on their preferred orientations (*Figure 6H*), which is consistent with the ISOI results (*Figure 1I*, *Figure 1—figure supplement 2A–D*). The average distances for within-group neuron pairs were shorter than the average distance between two randomly selected neurons (*Figure 6J*). Periodicity analysis shows the size of curve-orientation subdomain was around 360 μm (*Figure 6—figure supplement 2*).

Circle-preferring neurons also showed some interesting features. *Figure 6K* plots average responses to different contours for circle-preferring neurons (red) and curve-orientation neurons (green). For each stimulus, neurons' maximum orientation-responses were averaged. For circle-preferring neurons, their responses increased approximately linearly with the length of the circle fragment. This was not the case for curve-orientation neurons. Also, their responses to closed shapes seemed to be proportional to the general similarity between the shape and the circle. As a

comparison, *Figure 6L* plotted their orientation-averaged responses. The two curves are similar to those in *Figure 6K*, except that the green curve becomes lower on the left side, reflecting the modulation of curve orientation on the responses of the curve-orientation-preferring neurons.

## Diversity of contour tuning for neurons in the curvature domains

So far, we have shown the common response properties for neurons within the curvature domains and their subdomains. To have a full picture of the neuronal constituents within these functional domains, we examined the diversity of neuronal tunings inside the curvature domains.

In *Figure 7*, we show single-neuron response matrices from three example two-photon sites, including a circle-preferring site (A), a curvature-orientation-preferring site (B) and a rectilinear-preferring site (C). The location of these sites can be found in *Figure 6G*. For each site, 10 example neurons are shown. For the circle-preferring neurons (A), they prefer circles much better than the half circles. Their responses patterns also exhibited certain diversity, including some neurons responded modestly to the 90°-angle stimuli (cell 8–10) while others did not. They could respond well (cells 1, 2, 3, 5, 7, 9) or modestly (cells 4, 6, 8, 10) to hexagons. For cells in curve-orientation-preferring site (*Figure 7B*), response diversity appeared even larger. Different neurons showed strong selectivity to the trefoil shapes (cells 1, 4), or particular 90° angles (cell 3). Different degrees of circle-preference were also apparent among these neighboring neurons.

The rectilinear site shown in *Figure 7C* was also located in an orientation domain that preferring the 45° orientation. Many cells responded well to the triangles and the 45°-oriented lines. Some cells also responded to the 45°-orientated ovals and flat curves, likely due to their general orientation appearance. However, neighboring neurons could respond strongly to one of these stimuli (e.g. cell 1 to oval, cells 2 and 3 to triangle, cell 4 to line) but much weaker to others. Some neurons (e.g. cell 9) exhibited a wider range of preferred shapes than their neighbors.

Thus, for neighboring neurons, although their general shape preferences were similar, which constitute the basis of the ISOI signals, their contour response patterns still vary from cell to cell, and some of the differences were substantial. The degrees of such diversity also seem to be related to the types of these sites in V4.

Shape-related responses of neurons in curvature domains, rectilinear domains, and those outside of these two were summarized in *Figure 8*. In the orientation-sorted average response matrices, curvature neurons showed much higher response amplitudes, and larger differences between curved and rectilinear responses, than the other two types of neurons. Nevertheless, these neurons also responded to rectilinear stimuli, for example the triangles, at about half of the response amplitude. Interestingly, neurons in rectilinear domains did not show prominent differences in their responses to rectilinear and curved stimuli. This is also consistent with the weak rectilinear domains in ISOI imaging (*Figure 1D*). *Figure 8D* further illustrates the differences among three regions, in which the size and color of the shapes are proportional to the response magnitudes of the first rows in the response matrices in A–C.

## Discussion

Using two complementary imaging methods, we found a novel functional structure in V4, but not in V1 or V2, for curved contour processing. The size of these domains (518 µm) was similar to that of the orientation domains in this area; moreover, they tended to occupy regions outside the orientation domains. Within these domains, neurons preferring different curve features (e.g. curve orientation) further clustered into smaller subdomains. Interestingly, subdomains were also found in color- (*Li et al., 2014*; *Ghose and Ts'o, 2017*; *Liu et al., 2020*), orientation- (*Ghose and Ts'o, 1997*; *Tanigawa et al., 2010*), and direction- (*Li et al., 2013*) specific regions in V4. These nested functional architectures are similar to that of category domains in IT, where large category domains (animate vs. inanimate) contain subcategory domains (e.g. for face and body parts) (*Liu et al., 2013*; *Grill-Spector and Weiner, 2014*). Curvature domains are probably vertical columns. However, two-photon imaging revealed considerable complexity underlying these functional organizations. First, individual neurons usually responded to a wide range of stimuli with different levels of response amplitudes. Thus, curvature domains are not 'curvature-only' domains. Second, neurons inside the curvature domains and those outside do not completely differ in their stimulus coverage, although their optimal stimuli are usually different. Third, for neighboring neurons within a functional domain,

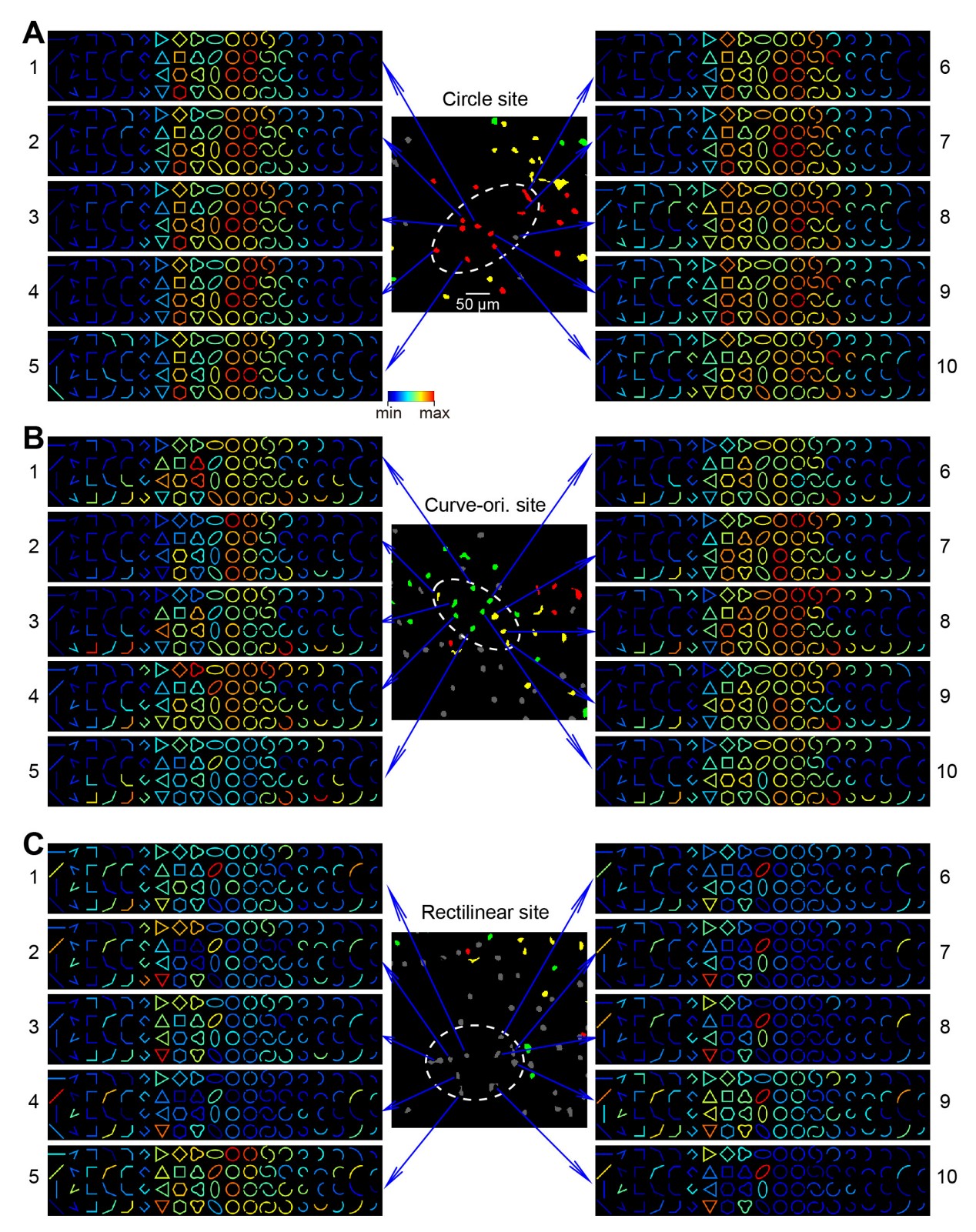

**Figure 7.** Single-neuron responses in three types of cortical sites. (A) Response matrices of 10 neighboring neurons in a circle-preferring two–photon site (also shown in *Figure 6G*). Each matrix is normalized to its maximal and minimal responses. (B) As in A, but for a curve-orientation-preferring site. (C) As in A and B, but for a rectilinear site.

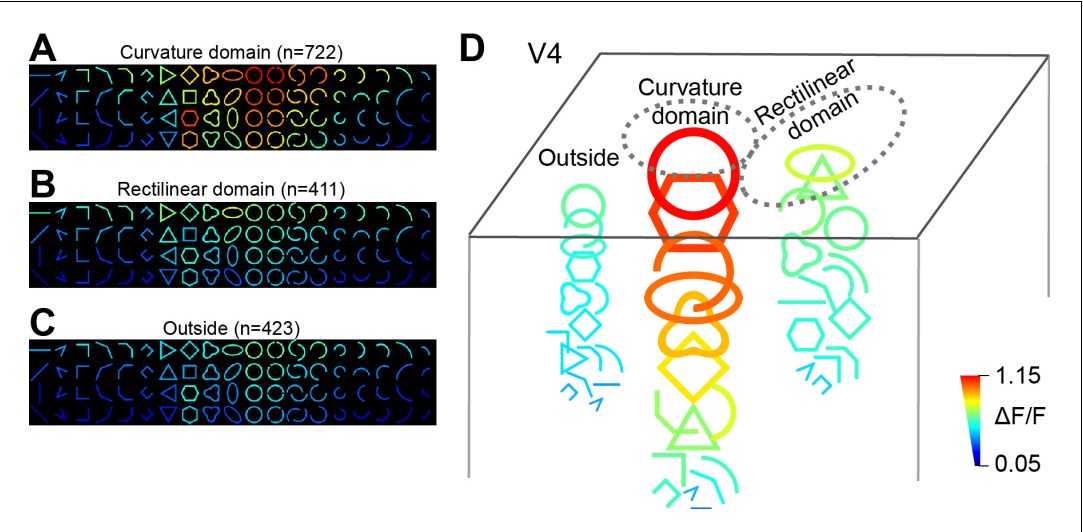

**Figure 8.** A summary of neurons' response patterns in different V4 regions. (**A–C**) Average response matrices (without normalization) from neurons in the curvature domains (**A**), rectilinear domains (**B**) and those outside of these two types of domains (**C**). Responses are orientation-sorted before averaging. (**D**) An illustration of the contour-preferences for neurons in the three regions, calculated based on the optimal-orientation responses in A–C (the top rows). The size and color of the shapes represent the strength of the corresponding responses. For clarity, not all stimulus icons were drawn.

although they showed common shape preferences, many of them exhibited different individual preferences, some of which were substantial. Thus, the functional maps we observed are best understood as a certain orderness of the complexity of shape responses in V4 population.

Consistent with previous findings (*Pasupathy and Connor, 1999*; *Pasupathy and Connor, 2001*; *Pasupathy and Connor, 2002*; *Yue et al., 2014*), we confirmed that V4 plays a key functional role in the representation and processing of curvature information. A majority of V4 neurons have been reported to exhibit complex multi-dimensional response patterns to shape stimuli (*Gallant et al., 1993*). Recent studies in V4 (*Bashivan et al., 2019*) and IT (*Ponce et al., 2019*) have shown that individual neurons in these areas can be best activated by non-natural synthetic patterns that are much more complex than those traditionally tested. Thus, classifying neurons into functionally separate groups based on their contour responses is often difficult. This is especially true when the testing stimulus set is limited (as in the present work). However, intrinsic functional structures in an area are suggestive of particular feature dimensions been emphasized in this area. We observed that V4 contains clear maps for curvature processing, which suggests that curvature is a unique feature dimension in this area, not only in single neurons but also as a populational feature. Further, both curved and rectilinear contours formed functional structures (curvature and orientation domains) with each functional structure containing nested subdomains for further detailed tuning (*Figure 6G and H*). This functional organization could underlie the parallel processing of curved and straight contours (*Ito, 2012*) and the different underlying pooling mechanisms (*Nandy et al., 2013*).

In human fMRI studies, V4 is more activated by circular stimulus than by curves (*Dumoulin and Hess, 2007*) or parallel patterns (*Wilkinson et al., 2000*). Using two-photon imaging, we observed many circle-preferring neurons (*Figure 6I*). However, these circle-preferring neurons also showed good responses to other non-circle stimuli (*Figure 6E*). Many curve-orientation-preferring neurons also responded best to circles (*Figure 6C*). This indicates that these circle-preferring neurons do not constitute a functionally distinct subdivision of V4.

Early single-cell recordings have suggested that V4 neurons may cluster based on their color or orientation preferences (e.g. *Tanaka et al., 1986*). Subsequent optical imaging studies confirmed these functional structures (*Ghose and Ts'o, 1997*; *Tanigawa et al., 2010*; *Li et al., 2014*). For shape-responsive neurons, clustering according to their selectivity to complex patterns has also been suggested (*Tanaka et al., 1986*; *Gallant et al., 1993*; *Gallant et al., 1996*), but studies have also shown that neurons selective to complex shape features are mixed with those selective to simple features such as orientations (*Kobatake and Tanaka, 1994*). Our results confirmed that,

although neurons tended to cluster according to their common response preferences, substantial diversity in contour tuning still exist among these clustered neurons. This heterogeneity may make these functional organizations less detectable in electrophysiological studies, for example, compared to the orientation columns in V1.

In a recent two-photon calcium imaging study posted on bioRxiv, *Jiang et al., 2019* imaged cellular responses in parafoveal V4 representations (~1° eccentricity from the fovea) to a large set of natural and artificial stimuli. Their findings suggested a separate coding for orientation, curve and angle. However, due to the spatial limitation of the two-photon technique, domain features and distributions are not available. With ISOI over a large imaging window (16–20 mm diameter), we observed the distribution of the curvature domains and their spatial relationships with other known functional maps (*Figure 2D*, *Figure 2—figure supplement 1*). We found that curvature domains existed in the entire exposed V4 surface (0–10° eccentricity from the fovea). Further, the ISOI maps of the curvature and orientation domains were comparable (*Figure 2F*), which suggests that they had a comparable degree of clustering.

In another study also submitted to eLife, *Hu et al., 2020* investigated curvature organization in area V4 with ISOI. Despite a different stimulus protocol they used, they observed curvature domains similar to what we observed. The major ISOI findings are similar between these two studies. With the power of two-photon imaging, we not only confirmed these ISOI findings but also revealed neuronal properties underlying these domains, including cellular-level preferences and the tuning diversity of neurons within these functional domains. Thus, *Hu et al., 2020* and this study both revealed the overall features of curvature domains in V4, while *Jiang et al., 2019* and this study both revealed cellular-level features of the functional domains. Only this study directly linked functional domain properties with their underling neural responses.

Curve-preferring neurons have been reported in V1, V2, and V4 (*Hegdé and Van Essen, 2007*; *Ponce et al., 2017*). However, we did not observe functional structures for curves in V1 and V2, which is consistent with a previous study (*Yue et al., 2014*). Curvature domains emerge in V4 (*Yue et al., 2014*; this study) and remain present in IT (*Yue et al., 2014*; *Srihasam et al., 2014*). Therefore, V4 contains both basic orientation (similar to V1 and V2) and curvature maps (similar to IT), which indicates its intermediate-stage role in the shape processing hierarchy. There is a progressive change in multiple feature dimensions along this hierarchy, for example receptive field size, response linearity, and degree of modulation by top-down controls. Functional architectures for shape processing form an independent dimension, which could contribute to a better understanding of the neural mechanisms for shape processing. Moreover, the consistency between functional architectures and shape processing levels highlights the importance of functional organization in the brain. This study provides another measurement type (domain-level activation) for studies on shape processing in addition to the cellular- and voxel-level measurements. Interesting topics, such as the functional importance of clustering, the plasticity of such clustering and its contribution to cognition, can be further explored.

We have shown that curvature neurons had smaller RFs than those of non-curvature neurons (*Figure 4L and M*). We further analyzed and showed that curvature neurons had stronger surround suppression (*Figure 4—figure supplement 1F and G*). Thus, consistent with previous findings, curvature neurons did exhibit end-stopping properties. However, curvature neurons showed additional curve-specific preferences, including curve-orientation tuning, preference for curves over similarly-sized angle stimuli, etc. For circle-preferring neurons, they prefer circles (larger) than the half-circles (smaller). These additional features suggest that curvature neurons were not simple end-stopping neurons, although they exhibited end-stopping properties.

Curvature domains are also unlikely the suppression-domains (S-domains) in V4 revealed by comparing activations to small and large patches of gratings (*Ghose and Ts'o, 1997*). In our study, curvature domains were revealed in difference maps, including: circles vs. triangles (*Figure 1D*), circles vs. bars (*Figure 1G*), curves vs. bars (*Figure 1H*), and curves vs. angles (*Figure 1—figure supplement 2L*). The two stimuli in each pair had similar sizes, and thus had similar surround suppression effects. Their contrast pattern in the difference maps could not be due to differences in suppression. The stimuli on the two sides also had balanced orientations, plus their maps had little overlap with the orientation maps (*Figure 2D*). The only consistent contrasts in these pairs of stimuli was the curvature feature. Thus, the curvature domains we found are unlikely related to suppression and orientation domains.

Our two-photon results have limitations in terms of the stimulus set, anesthetized preparation, and calcium signals. We only tested 19 simple contour shapes, mostly with four orientations. Considering the complexity of single-cell response patterns we observed, the best-driven stimuli we identified may not be the optimal ones for the neurons. Although a more in-depth analysis of the current data set can still improve our estimation of the optimal stimuli, new stimuli, or experimental approaches seem to be necessary. A recently reported deep-learning-network approach successfully found optimal stimuli for V4 neurons (*Bashivan et al., 2019*). The optimal stimuli they found were usually complex patterns and unlike the contour shapes previously tested. However, these optimal stimulus patterns also contained specific curvature features similar to the optimal ones obtained with a parametric stimulus set like ours. Thus, it is likely that the best-driven contour features we obtained should also be represented in the actual optimal stimuli for the neurons we measured. Our ISOI tests also provided a practical way in designing and narrowing down the stimulus set for the subsequent targeted two-photon imaging.

The primary new finding of this study is the functional architecture for curvature processing in V4. The curvature domain likely shares similar formational rules and functional significances with other functional architectures previously discovered (*Mountcastle, 1997*; *Chklovskii and Koulakov, 2004*). The significance of this particular structure is based on several observations described in Results: First, the curvature domains were found in V4 but not in V1 or V2, although curvature neurons were also found in the latter two areas (*Hegdé and Van Essen, 2007*; *Ponce et al., 2017*). Second, the domain strength was higher for V4 curvature domains than those for angles. These two factors are consistent with the notion that V4 plays an important role in curvature processing. Third, features of these domains are helpful in understanding the underlying neural processes. For example, the relative separation of the curvature and rectilinear domains suggests a parallel processing of these two contour features. The existence of the circle and curve-orientation sub-domains within the curvature domains suggests that there are different processes for curvatures. Finally, the existence of curvature domains also provides an efficient way to study curvature neurons in V4, for example, by targeted recording, imaging, or tracer labeling.

## Materials and methods

### Key resources table

| Reagent type (species) or resource | Designation | Source or reference | Identifiers | Additional information |
|---|---|---|---|---|
| Strain, strain background (Macaque, male) | Macaca mulatta | Suzhou Xishan Zhongke animal Company, Ltd Hubei Topgene Biotechnology Co.,Ltd | | http://xsdw.bioon.com.cn/ http://topgenebio.com/ |
| Strain, strain background (Macaque, male) | Macaca fascicularis | Beijing Inst. of Xieerxin Bology Resource | | http://www.xexbio.com/ |
| Recombinant DNA reagent | AAV1.Syn.GCaMP6S. WPRE.SV40 | Addgene | v25497 | |
| Recombinant DNA reagent | AAV9.Syn.GCaMP6S. WPRE.SV40 | Addgene | CS1282 | |
| Software, algorithm | MATALAB R2017b | MathWorks | | https://www.mathworks.com |
| Software, algorithm | Codes for ISOI data analysis | This paper | | https://osf.io/qydj5/ |
| Software, algorithm | Codes for 2P data analysis | This paper | | https://osf.io/qydj5/ |

A total of seven hemispheres (i.e. cases) from six adult male macaque monkeys (five *Macaca mulatta*, one *Macaca fascicularis*) were imaged. ISOI were performed on all seven cases, two-photon imaging were performed on two cases (Case 3 and Case 4). These monkeys also participated in other studies. All procedures were performed in accordance with the National Institutes of Health Guidelines and were approved by the Institutional Animal Care and Use Committee of the Beijing Normal University (protocol number: IACUC(BNU)-NKCNL2016-06).

## Surgery procedures

Animals were sedated with ketamine (10 mg/kg) or Zoletil (tiletamine HCl and zolazepam HCl, 4 mg/kg) and transferred to the lab. They were artificially ventilated on a stereotaxic and anesthetized with isoflurane (1.5–2.5%) during surgery. A 22–24 mm (diameter) circular craniotomy and durotomy were performed (center location, 30–37 mm from midline, 15–24 mm from posterior bone ridge) to expose visual areas V1, V2, and V4 (illustrated in *Figure 1A*). ISOI was performed right after the surgery (see below). Then, one of two types of protocols was chosen: For cases only used for ISOI, we implanted an optical chamber and recovered the animal. For cases used for two-photon calcium imaging, we injected virus into the visual cortex, placed back the bone and sealed the craniotomy (described below). After 1.5 months, we reopen the craniotomy and implanted an optical chamber for the following weekly based two-photon imaging experiments. The same type of chronic chamber was used for ISOI and two-photon imaging (*Li et al., 2017*). The inside diameter of the chamber was 13–16 mm, thickness of the glass is 0.18 mm. Case 2 was only imaged once and then used for other studies thus no chamber was implanted.

## Intrinsic signal optical imaging (ISOI)

ISOI were performed either right after the surgery (without a chamber), or on a weekly-base after a chamber was implanted. Imaging usually lasted for 8 hr. Right before the imaging, anesthesia was switched from isoflurane to a mixture of propofol (induction 2 mg/kg, maintenance 2 mg/kg/hr) and sufentanil (induction 0.15 µg/kg, maintenance: 0.15 µg/kg/hr) (in four cases), or to Zoletil (tiletamine HCl and zolazepam HCl, induction 4 mg/kg, maintenance 1.25 mg/kg/hr, in three cases). The monkeys were immobilized with vecuronium bromide (induction 0.25 mg/kg, maintenance 0.05 mg/kg/hr) to prevent eye movements. Anesthetic depth was assessed continuously via monitoring the electrocardiogram, end-tidal $CO_2$, blood oximetry, and body temperature. Eyes were dilated (atropine sulfate, 0.5 mg/ml) and fit with contact lenses of appropriate curvatures to focus on a stimulus screen 57 cm from the eyes.

Images of reflectance change (intrinsic hemodynamic signals) corresponding to local cortical activity were acquired (Imager 3001, Optical Imaging Inc) with 632 nm illumination. Frame size was 540 $\times$ 654 pixels, representing either 15.5 $\times$ 19 mm or 18 $\times$ 22 mm of imaged area, depending on the lenses chosen. For each trial, imaging started 0.5 s before the stimulus onset and collected at 4 Hz frame rate. Each visual stimulus was presented for 3.5 s. The total imaging time for each trial was 4 s, during which 16 frames were imaged. Interstimulus intervals were at least 6 s. Stimulus conditions were displayed in a randomized order.

## Virus injection

Virus injection and two-photon imaging procedures were similar to those described in *Li et al., 2017*. In two cases (Case 3 and Case 4), we injected 500 nL AAV9.Syn.GCaMP6S.WPRE.SV40 (CS1282, titer 3.34e13 GC/ml, Addgene), or AAV1.Syn.GCaMP6S.WPRE.SV40 (v25497, titer 2.5e13 GC/ml, Addgene) into 10–15 cortical locations at a depth of 500 µm. After virus injection, cortex was covered with a piece of artificial dura. And the remaining dura was covered and glued to the top of the artificial dura with medical adhesive (Beijing Compont medical devices Co. Ltd). The original bone was placed back, secured with titanium lugs and bone wax in the gap. The scalp was sutured. A second surgery was performed 1.5 months later, in which the old craniotomy was reopened and an optical chamber was implanted for the following ISOI and two-photon imaging.

## Two-photon imaging

Two-photon calcium imaging was performed on a weekly-base after the chamber implanting. Animal anesthesia and preparation were the same as those in the ISOI experiments. Two-photon microscope was a Brucker Ultima IV Extended Reach (Bruker Nano Inc). Laser was generated with a Chameleon Ultra II (Coherent Inc). The excitation wavelength of the laser was set at 980 nm. Scanning frame rate was 1.3 Hz in a galvo scanning mode. Under a 16X objective (0.8 N.A., Nikon), 515 $\times$ 512 pixel images were collected, representing a 830 $\times$ 830 µm cortical surface. Imaging was continuous and the beginning of each stimulus presentation was synchronized with the beginning of a frame scanning.

As the microscope was vertical, we rotated the stereotaxic and the animal for ~45° so that the chamber plane was perpendicular to the laser beam. We imaged 10 cortical locations in the two virus-injected chambers (*Figure 3—figure supplement 1A and E*). Images were normally collected at a depth between 210 and 350 μm from the cortical surface. In six locations, we imaged at two depths, a depth around 230 μm and a depth around 310 μm (*Figure 3—figure supplement 1D and H*). During the imaging session, slow drifts of cortex in the imaging window was observed. The drift was normally less than 50 μm in the X-Y plane and less than 150 μm along the Z-axis in a course of 6–8 hr. We closely monitored the cell features during the imaging and adjusted position of the focal plane accordingly. Drifts in the X-Y plane were further corrected in offline data analysis (described below). In addition, population receptive field locations were re-plotted every 1–1.5 hr in order to check the stableness of the eye positions. In half of the experiments, some eye drift (1–2°) was detected during the imaging session. If drift was larger than 1°, we repeated the last stimulus run at the new receptive field position. If the drift was smaller than 1°, we adjusted the stimulus location for the following stimuli and continued the imaging. Replication of two-photon imaging of the same neuron population were tried but was not systematically tested. A general replication of the population-level responses was qualitatively observed.

## Visual stimuli for ISOI

Visual stimuli were generated using ViSaGe (Cambridge Research Systems Ltd.) and presented on a CRT monitor positioned 57 cm from the eyes. The stimulus screen was gamma corrected and worked at 100 Hz refreshing rate. We compared contour preference maps obtained with monocular and binocular conditions in Case 1 (*Figure 1—figure supplement 2M and N*), as well as two monocular conditions in Case 4 and did not find obvious differences. Since ISOI signal is normally stronger in binocular condition, all but one case (Case 5) were imaged in binocular conditions. For orientation and color preference maps, full-screen drifting sinewave gratings were used. The mean luminance was 28.9 cd/m$^2$. Gratings of two SF (0.25 and 1 c/deg, except for Case 6 in which only 0.25 c/deg were tested) and two (45° and 135°, for Cases 5 and 6) or four orientations (0°, 45°, 90°, and 135°, for other five cases) were tested. Gratings were drifting at 4°/s along a random direction perpendicular to its orientation and were presented in a random order. The initial phases of the gratings were also randomly selected.

For contour shape preference maps (e.g. *Figure 1D–L*), drifting contour patterns were used. Each contour element was ~2.5° and placed in a 3 × 3° grid (*Figure 1C*). The white contour lines was 0.2° in width and had a luminance of 111.2 cd/m$^2$. The dark background had a luminance of 20.6 cd/m$^2$. For each trial, a contour pattern was presented and drifted at one out of four or eight directions for 3.5 s at a speed of 4°/s. Interstimulus intervals were 6 s, during which a gray screen (20.6 cd/m$^2$) was presented. The initial phase (i.e. relative position) of the pattern was random. Each stimulus was repeated for 25–50 times. The essential stimuli (e.g. circle, triangle, gratings) was obtained in all seven cases (*Figure 2—figure supplement 1*), new stimuli were added in later cases and are described in the Results whenever is necessary.

## Visual stimuli for two-photon imaging

Visual stimuli used for two-photon imaging were generated in a similar way as those in the ISOI imaging and presented on a 21-inch LCD monitor (Dell E1913Sf) positioned 57 cm from the eyes. The stimulus screen was gamma corrected and worked at 60 Hz refreshing rate. All stimuli were bright stimulus (80.8 cd/m$^2$) on a gray (13.2 cd/m$^2$) background and presented monocularly to the left eye.

Population RF location was first mapped manually with a 2–3° circular patch of square-wave gratings, during which cortical fluorescent response was monitored. Then the RF were systematically mapped at the potential region with a single 2–3° patch of stimulus presented on a grid of 5 × 5 locations. Each position was tested with a square-wave grating, a random dot patch, and a circle, each moved at one out of four directions (45°, 135°, 225°, 315°) at 4°/s speed. Each stimulus was presented for 1.5 s. Interstimulus interval was 0.5 s. Population RF center (normally 5–7 degrees eccentricity) was then analyzed online based on cortical responses to these stimuli.

To obtain detailed RF size information, we presented circular patches of square-wave gratings (SF = 1 c/degrees, TF = 4 c/s, duty cycle = 0.2) of six different sizes (0.5°, 1°, 2°, 4°, 8°, 12°) at the

population RF center mapped before. Each stimulus was drifted along one of eight directions, and presented for 2 s with a 3 s interval. Each stimulus was repeated for seven times. From these stimuli, a size-tuning curve was calculated, which normally peaked at patch sizes of 2°–3.5°. Population RF size was then defined as twice of this size (i.e. 4°–7°).

To test the contour-shape preferences of the neurons, we presented single contour elements in the population RF. Single contour element was the same as those used in the ISOI experiments. The size of the contour was adjusted to 40% of the population RF. Each stimulus was presented for 2 s, during which it first appeared in the center of the population RF and drifted along one of eight directions (randomly selected). After it moved to the edge of a virtual window equal to the population RF diameter, it disappeared and reappeared in the opposite side of the window, and continue its motion along the same direction (illustrated in *Figure 3—video 1*). The speed of the contour movement was adjusted to (half of the diameter of the population RF)/s. The interstimulus intervals were 3 s. Each stimulus was usually repeated for 10–15 times. As shown in *Figures 6A*, 19 different contour shapes were used, most of them was presented in four orientations, two stimuli were presented in two orientations (square, hexagon), and one stimulus (half circle) was presented in eight orientations. This made up 73 different stimulus conditions. We also included four identical circles instead of 1 to be comparable to other stimuli's four orientations. The final stimulus matrix contained 76 stimuli.

## ISOI data analysis

ISOI data analysis was performed using MatLab 2017 (The MathWorks, Natick, MA). We first obtained $\Delta R/R$ response to each stimulus using following formula $\Delta R/R = (R_{8-16} - R_{1-4})/R_{1-4}$, in which $R_{8-16}$ is the average of frames 8–16, $R_{1-4}$ is the average of frames 1–4. The $\Delta R/R$ frames for each trial were then used for following analysis.

We used a pattern classifier (support vector machine, SVM) for calculating contrast maps between $\Delta R/R$ frames of two stimulus conditions. Compared with simple subtraction maps, SVM weight map achieved a higher signal-noise ratio by suppressing blood vessel noise and low-frequency noises (*Xiao et al., 2008*; *Chen et al., 2016*). The Matlab SVM program was provided by Chih-Jen Lin (LIB-LINEAR: A Library for Large Linear Classification, 2008; available at https://www.csie.ntu.edu.tw/~cjlin/liblinear/).

For each SVM map (e.g. *Figure 1D–L*), it was first Gaussian low-pass filtered (size = 5 pixels, std = 1), then clipped at $0 \pm 8$ SD for display, in which SD was the standard deviation of the background pixels values (blood vessel pixels and outside-chamber pixels, determined based on the blood vessel maps like the example shown in *Figure 1B*). For display purposes, the original map size (540 × 654 pixels) was cropped to 540 × 540 pixels by trimming off some background regions on the two edges.

To separate domain region from non-domain regions (e.g. *Figure 2D*), we used $0 \pm 2$ SD as the threshold levels and discarded resulting regions containing less than 50 pixels. Curvature domains were identified as the dark regions in the circle vs. triangle maps. Domain coverage in *Figure 2D* and the neuron classification in *Figures 5–8* were all based on this method. Only in estimating domain sizes (*Figure 2E*), we used a watershed method in order to deal with the 'connected domain' problem (see below). Orientation domains were identified as both the dark and white regions in the orientation maps. Color domains were identified as both the dark and white regions in the color vs. luminance maps. To compare with two-photon results (e.g. *Figure 4J and K*), regions in the SVM map that aligned with the two-photon imaging area were extracted and resized (bicubic interpolation) to 512 × 512 pixels for further analysis.

To estimate domain sizes (e.g. *Figure 2E*), neighboring domains were often connected and we separated them using a watershed algorithm. First, the image values were reversed to make the negative domains positive. Then we located the strongest response pixel, and cut out a 31 × 31 (0.9 × 0.9 mm) pixel region around this pixel. For all pixels in this square that had a value lower than 0 +2SD, they were set as 0+2SD. The response pattern was fitted with a two-dimentional Gaussian. An oval was obtained from the two-dimentional Gaussian with a threshold so that the oval had half of its pixels valued larger than 0+2SD (and the other half pixel all had values of 0+2SD). This oval was taken as the representation of this curvature domain. Before search for the next domain, the pixels in the oval region in the original map was set as 0+2SD. The searching stopped until no more pixels had value larger than 0+2SD. In addition, small domains had fewer than 75 pixels were not

included in the size analysis. Individual domain size was described as the diameter of an equivalent disk. Domain sizes were then averaged according to map types and cases. To measure the map strength of a functional map (e.g. *Figure 2F*), we calculated the differences of average pixel values in white and dark domains in the subtraction map, using the domain masks described above. For example, the map strength of a 45° vs. 135° orientation map was calculated as the subtraction between the average pixel values of 135° domains and the average pixel values of 45° domains in the 45°−135° subtraction map.

## Two-photon data analysis

Two-photon data analysis was performed using MatLab 2017 (The MathWorks, Natick, MA). Image alignment and cell identification algorithms were modified from those used in *Li et al., 2017*. To correct slow cortex drifts and biological movements (likely due to heart pulsation and artificial ventilation) within the X-Y plane (normally less than 20 pixels during the whole imaging session, typically five pixels), we first obtained a frame template by averaging 1000 frames from a chosen session, then all frames of that experiment, and frames from other days of the same cortical layer, were aligned to this template using a cross-correlation method.

Fluorescent images (e.g. *Figure 3C*) were obtained by averaging all collected two-photon images during the imaging session, including the baseline and response images. Response images (e.g. *Figure 3D and E*) were obtained by subtracting the baseline image (average of the two frames before the stimulus onset) from the response image (average of the second and third frames after the stimulus onset).

Cell bodies were first identified based on their responses to each visual stimulus. We first averaged all repeats for a stimulus, then obtained a subtraction image by subtracting the baseline image (average of 2 immediate pre-stimulus frames) from the response image (average of frames 2 and 3 after the stimulus onset). This subtraction image was then filtered using two separate Gaussian filters: G1 (size = 7 pixels, std = 1.5) and G2 (size = 15 pixels, std = 7). Then the subtraction map of G1-G2 was used for cell extraction. The cells extracted in this way had smooth edges and filled cell-body. For each cell, the cell body identified from its best-responsive stimulus was used as its cell body and was used to calculate its responses in all the other stimuli.

A cell was identified if more than 30 connected pixels had values larger than mean+2.75SD, in which SD is the standard deviation of the pixel values of the whole filtered subtraction image. After cell locations were determined, we calculated fluorescence responses for cells to each stimulus using following formula: $\Delta F/F0 = (F-F0)/F0$, in which F is the average pixel value covered by the cell in the response image, and F0 is average pixel values in the baseline image. A cell was included for the following analysis if any of its stimulus responses ($\Delta F/F0$) was larger than 0.3 for the stimuli tested.

## Orientation tuning to curves

A cell was identified as a curve-orientation-preferred cell (e.g. *Figure 6A–D*) if its responses to eight curve orientations passed the Rayleigh test for circular uniformity (*Fisher, 1993*) at a significant level of p<0.05. Then we fitted the responses with a modified von Mises function (*Mardia, 1972*):

$$F(\theta) = a0 + b1 \times e^{c1 \times \cos(\theta - \theta p)} + b2 \times e^{c2 \times \cos(\theta - \theta p - 180)}$$

where θ is the orientation of curve (0–360°); θp is the preferred orientation; a0 is the baseline offset; (b1, b2) and (c1, c2) determine the amplitude and shape of the tuning curve, respectively. Fitting parameters were obtained with a least-square nonlinear regression method (nlinfit in Matlab, Mathworks). Goodness of fit ($R^2$) values were >0.7 for all neurons which have significant orientation tuning to curve (n = 417). Cells' preferred orientations were then calculated from the fitting curve (*Figure 6B*).

## Circle preference

A cell was determined as circle-preferred cell if it met following two criteria: 1. The cell's response was significantly modulated by contour types (bar, 90°-angle, triangle, curve, each was represented by its optimal oriented stimulus, and circle, p<0.05, one-way ANOVA) and its maximum response was to the circle. 2. The cell's response to the circle was significantly larger than its response to the curve of its preferred orientation (p<0.05, paired t-test).

### Shape preference

In calculation of the preferential responses to two contours (e.g. *Figure 4J and K*), we first normalized the cell's ΔF/F0 responses to the whole stimulus set to 0–1, then the differences of the responses to the two types of stimuli were calculated.

### RF size and surround suppression

To measure cell RF sizes (e.g. *Figure 4L and M*), we first performed a two-way ANOVA analysis (with repetition) for the cell's responses to square-wave gratings of 6 different sizes and eight different moving directions. If the size factor is significant ($p < 0.05$), then we fitted a ratio of Gaussian function (*Cavanaugh et al., 2002*) to the direction-averaged size tuning curve:

$$\mathrm{R(x)} = \frac{k_c L_c(x)}{1 + k_s L_s(x)}$$

$$L_c(x) = \left( \frac{2}{\sqrt{\pi}} \int_0^x e^{-(y/wc)^2} dy \right)^2$$

$$L_s(x) = \left( \frac{2}{\sqrt{\pi}} \int_0^x e^{-(y/ws)^2} dy \right)^2$$

where x is the stimulus diameter, $k_c$ and $k_s$ are gains for the RF center and surround, $L_c$ and $L_s$ are the total squared responses of the center and surround, and wc and ws are the size of the center and surround (wc <ws). Fitting parameters were obtained with a least-square nonlinear regression method (nlinfit in Matlab, Mathworks). The optimal stimulus size was determined as the diameter corresponding to the peak response in the fitted function. Goodness of fit ($R^2$) values were >0.7 for all neurons that had significant size tuning to square wave gratings (n = 1493), and 95% (1418/1493) of these neurons had a preferred stimulus size between 0.5° and 12°. If a cell's responses were not modulated by stimulus size (i.e. size factor was not significant in the two-way ANOVA analysis), then this cell was not included in the RF size related analysis. Surround suppression index (e.g. *Figure 4—figure supplement 1F*) was calculated from the size tuning curve as (maximum response – minimum response)/maximum response, in which minimum response was the least response obtained for stimulus size larger than that of maximum response (maximum surround suppression).

### Pixel-based contour-preference map

Pixel-based contour-preference maps (e.g. *Figure 4B*) were a combination of a gray value (G) map and a red/green hue map. For each pixel, we first calculated a shape preference index: SPI = abs (ΔFcc-ΔFbat)/max(ΔFcc,ΔFbat), where ΔFcc is the maximum fluorescence increase to curves and circles (red framed ones in *Figure 3F*), ΔFbat is the maximum fluorescence increase to bar, angle and triangle (green framed ones in *Figure 3F*). Then we calculated the gray value as: G = 200xSPIxFp/Fp_f, where Fp is the maximum of the fluorescence values in the response images to all five types of stimuli, Fp_f is the average of all pixels' Fp in the same frame. Hue for a pixel is red if its ΔFcc is larger than ΔFbat, and is green if not.

### PCA

PCA results were shown in *Figure 5* and *Figure 6—figure supplement 1*. For each neuron, its response matrix was normalized to 0–1. Its responses to different orientations were sorted, so that for each contour, 0° represented its highest response and 135° for the lowest (thus no longer correspond to the stimulus icons). Only neurons tested with the full stimulus set were analyzed (n = 1556). The responses of these neurons were arranged as a 2D matrix, and the mean response to each stimulus was subtracted before the PCA analysis was performed. In *Figure 5E*, the color of the neurons was determined based on their coordinates on PC1.

### Statistical analysis

All t-tests and paired t-test are two-tailed.

## Acknowledgements

This work was supported by the National Natural Science Foundation of China (31530029 and 31625012 to HDL and 31800870 to RT) and the China Postdoctoral Science Foundation (2018M631373 to RT). We thank Dr. Shiming Tang for valuable technical support on two-photon imaging. Lab members Jie Lu, Yan Xiao, Chen Fang, Kun Yan, Jingting Xu, Heng Ma, Jiayu Wang, Pengcheng Li, Chen Liang, and Wenhao Zhao provided technical assistance.

## Additional information

### Funding

| Funder | Grant reference number | Author |
|---|---|---|
| National Natural Science Foundation of China | 31530029 | Haidong D Lu |
| National Natural Science Foundation of China | 31625012 | Haidong D Lu |
| National Natural Science Foundation of China | 31800870 | Rendong Tang |
| China Postdoctoral Science Foundation | 2018M631373 | Rendong Tang |

The funders had no role in study design, data collection and interpretation, or the decision to submit the work for publication.

### Author contributions

Rendong Tang, Conceptualization, Resources, Data curation, Software, Formal analysis, Supervision, Funding acquisition, Validation, Investigation, Visualization, Methodology, Writing - original draft, Project administration, Writing - review and editing; Qianling Song, Conceptualization, Resources, Data curation, Software, Formal analysis, Supervision, Validation, Investigation, Methodology, Project administration; Ying Li, Rui Zhang, Xingya Cai, Data curation, Investigation; Haidong D Lu, Conceptualization, Resources, Data curation, Supervision, Funding acquisition, Validation, Investigation, Methodology, Writing - original draft, Project administration, Writing - review and editing

### Author ORCIDs

Rendong Tang https://orcid.org/0000-0002-3622-3383
Qianling Song http://orcid.org/0000-0001-9177-7429
Xingya Cai http://orcid.org/0000-0001-7829-3833
Haidong D Lu https://orcid.org/0000-0003-1739-9508

### Ethics

Animal experimentation: All procedures were performed in accordance with the National Institutes of Health Guidelines and were approved by the Institutional Animal Care and Use Committee of the Beijing Normal University. Protocol number: IACUC(BNU)-NKCNL2016-06.

### Decision letter and Author response

Decision letter https://doi.org/10.7554/eLife.57502.sa1
Author response https://doi.org/10.7554/eLife.57502.sa2

## Additional files

### Supplementary files

- Transparent reporting form

## Data availability

Data and MATLAB code required to reproduce all figures are available at https://osf.io/qydj5/.

The following dataset was generated:

| Author(s) | Year | Dataset title | Dataset URL | Database and Identifier |
|---|---|---|---|---|
| Tang R, Song Q, Li Y, Zhang R, Cai X, Lu HD | 2020 | Source data and codes for V4 manuscript | https://osf.io/qydj5/ | Open Science Framework, 10.17605/OSF.IO/QYDJ5 |

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
