## [Decision Letter]

**Acceptance summary:**

Curvature is a fundamental element of complex visual shapes in the world around us. This study elegantly combines cutting edge methodologies to investigate – from single brains cells to functional domains – how V4, a mid-level area of the visual cortex of primates, shows specialisation for processing and representing curvature, not seen at earlier levels of visual processing. This functional architecture complements known domains for orientation tuning and colour in V4 and significantly furthers our understanding of how primate brains recognise and discriminate visual objects.

**Decision letter after peer review:**

Thank you very much for submitting your article "Curvature-processing domains in primate V4" for consideration by *eLife*. Your article has been reviewed by three peer reviewers, and the evaluation has been overseen by a Reviewing Editor and Joshua Gold as the Senior Editor. The following individuals involved in review of your submission have agreed to reveal their identity: Ed Connor (Reviewer #1); Aniruddha Das (Reviewer #2).

The reviewers have discussed the reviews with one another and the Reviewing Editor has drafted this decision to help you prepare a revised submission.

As the editors have judged that your manuscript is of interest, but as described below that additional data analyses are required before it is published, we would like to draw your attention to changes in our revision policy that we have made in response to COVID-19 (https://elifesciences.org/articles/57162). First, because many researchers have temporarily lost access to the labs, we will give authors as much time as they need to submit revised manuscripts. We are also offering, if you choose, to post the manuscript to bioRxiv (if it is not already there) along with this decision letter and a formal designation that the manuscript is "in revision at *eLife*". Please let us know if you would like to pursue this option. (If your work is more suitable for medRxiv, you will need to post the preprint yourself, as the mechanisms for us to do so are still in development.)

Summary:

Lu and colleagues present potentially compelling new results on the architecture of curvature response strength and selectivity in macaque visual cortical area V4, in comparison to the lack thereof in earlier level visual areas V1 and V2. A particular strength is that the authors combine intrinsic-signal optical imaging (giving meso-scale resolution over a few mm of cortex) with 2-photon imaging (single cell resolution, over a few 100 microns; including 2-photon imaging at multiple cortical depths). The intrinsic signal data show distinct clustering of responsiveness to object fragments that constitute "intermediate" visual representations that are more complicated that the traditional single orientation columns in V1 or V2. However, the reviewers raised a number of critical issues relating to analysis, presentation and interpretation of this exciting data set and hope that these can be satisfactorily addressed in order for the study to deliver its full potential impact.

Essential revisions:

1) To include single activation condition maps for the intrinsic optical imaging and explore whether map activations could be explained by simpler, underlying response patterns.

The presented intrinsic optical imaging data comprise only difference maps, which could potentially obscure underlying results. For example, in Figure 1L, it could be that those 2 angles elicit very little localized responses, or that they elicit an identical pattern of activation, and from the difference map we don't know which is the case. It would be good to see some single condition activation maps.

That would also potentially help in investigating to what extent these responses can be explained by simple receptive field (RF) summation. For instance, there is a substantial literature in associating preferential responses to curves with end-stopped neurons: Hubel and Wiesel, in 1965 suggested that end-stopped, length tuned neurons could be useful for curvature detection and there was a nice explicit model of how that could happen by Dobbins et al. in 1987. But the manuscript makes little reference to any size tuning or surround suppression, and how that, in combination with classic orientation tuning, could create a cell that responds vigorously to curvature. This is even a greater concern given the well-established prevalence of size tuning, both electrophysiologically (1987) and with optical imaging (1997), in area V4. This has a huge impact on both novelty and interpretability; if you have a region of surround suppression (which we know exist) and it overlaps with an orientation region, this can look like a "curvature" region in that it will respond to a curve better than extended straight line or grating. But we would argue it's not a "curvature" region, since a short bar or grating without any curvature is actually the optimal stimulus. For example, can the response to triangles, on a pixel by pixel basis, be explained by simply adding or averaging the response the constituent orientations? Similarly, can the circle responses be explained by adding the responses to the half-circles?

2) To support the 2-photon results with convincing anatomical images.

One technical issue is the lack of any good underlying anatomical images to support the 2-photon data. This is concerning because all the images provided here show cells that are filled in, but living cells should appear as rings representing the cell membrane. Filled in cells are dead or compromised, and this would be a critical caveat for all the response results reported here. All of the cells in Figure 3C-F appear to be filled in. This could be because (i) these images do not have sufficient resolution to show rings, (ii) the authors did some kind of smoothing of these images (smoothing is mentioned in Materials and methods), or (iii) the cells are filled with calcium, reflecting a dead or compromised state. This concern is further exacerbated by the strange cell shapes in Figures 4G and H and 5A, D, and G. The authors should show the underlying anatomical images for those panels and explain how they defined the odd shapes shown in them. In addition to showing anatomical images, the authors should quantify how many cells are filled or ring-like in their images, and clarify whether the cells for which they report tuning include filled cells (and how many).

3) The reviewers would like to see more systematic and consistent analysis and representation across Figures 4. and 5 that allows the reader to better understand the fundamental, underlying tuning properties of neurons.

A technical concern is that Figure 4 and Figure 5 appear to present the same phenomena, but with different analyses for different images. Figure 4G labels neurons according to whether they exhibited stronger responses to circles or arcs. Figure 5 distinguishes neurons by how closely they are correlated with the average response pattern for the local curvature domain. But the examples in Figures 5B,C,E,F,H,I make it clear that the differences between correlated and not correlated again have to do with relative responses to circles vs. arcs. All of these images in Figures 4 and 5 need to be analyzed in the same ways, with an emphasis on the more explanatory analyses related to tuning, rather than unexplained correlation.

In relations to the tuning properties underlying the impressive 2-photon data, the question arises to what extent can RF summation (for example, RF summation models that include non-linearities that have been applied to V4) explain responses across the stimulus array?

See also point 4 below, in relation to this issue of the underlying tuning properties of neurons.

4) To reanalyze both datasets (intrinsic optical imaging and 2-Photon) in light of what is well-known about V4 tuning for fragments, not whole shapes, for curvature acuteness (which means responses to triangles reflect high curvature acuteness tuning) and object-relative position of contour fragments (which explains the differences between "circle" neurons and "curve" neurons, as well as the difference between triangle responses and angle responses, most of which we do not get to see). This should radically change the interpretation of Figures 1 and 2, and change the analysis emphasis of regions in the 2-photon section of the paper.

The major technical and interpretational concern is that differences between response patterns throughout the paper are presented as though the neurons were tuned for circles in some cases, arcs in others, triangles in others, etc. This ignores well-established tuning dimensions in V4 that explain response patterns like these at a more basic level. Pasupathy and Connor, 1999; 2001and 2002) and Carlson et al. (2011, Current Biology) clearly demonstrated that:

i) V4 neurons respond to contour fragments, not complete shapes,

ii) V4 neurons are strongly tuned for the object-relative position of those contour fragments,

iii) V4 neurons are tuned for curvature acuteness, with a bias toward sharp curvatures,

iv) V4 neurons are differentially tuned for convexity and concavity.

By ignoring this literature, the authors might misinterpret many of their results:

a) In Figures 1 and 2, responses to triangles are treated as representative of non-curvature regions. But, per point (iii) above, the points of those triangles will drive strong responses from the most acute curvature tuning regions. Thus, the curvature/triangle contrasts are more likely to represent organization for broad curvature vs. acute curvature (angles are geometrically just the limiting case of acute curvature) than a difference between curvature and straight lines. The same is true for line segments, whose terminations drive weaker but similar responses, and which show only weak differences from triangles and angles. The only valid contrast for finding curvature domains is curves vs. gratings, which reveals very different regions than triangles in Figure 1. This reliance on circle vs. triangle contrast has ramifications throughout the paper, because it is used to define curvature domains and thus bias the 2-photon studies away from regions of acute curvature tuning.

b) A similar problem appears in Figure 3, where the only response shown for the "rectilinear" region is a highly selective response for a triangle. Per point (ii) above, triangles provide a whole object context in which V4 neurons can exhibit their strong tuning for object-relative position of contour fragments, including high-curvature angles. This is why neuron 3 does not respond to any other rectilinear stimulus. If triangle responses like this are the basis for the green regions in Figure 3, then those regions are acute curvature regions, not non-curvature regions, and again the authors will be excluding high-curvature regions from their analyses in Figures 4 and 5.

c) Per above, Figures 4 and 5 are restricted to clusters of broader curvature tuning, strongly biasing the analyses. This is undoubtedly one reason that many example neurons respond strongly to circles. The other major problem with these figures is that they are presented as evidence that many neurons in V4 are strongly selective for circles as whole stimuli, while others are strongly selective for arcs over circles. In fact, the references cited above make clear that neurons in V4 are invariably tuned for shape fragments (i), not whole shapes, and they are strongly tuned for fragment position within shapes (ii). This is why many neurons respond most strongly to circles, because they are tuned for an arc-shaped fragment, but respond much more strongly when that fragment is in the preferred object-relative position. This is frequently true for circles, because there is also a strong correlation between curvature orientation tuning and object position tuning, since convex curves pointing up usually occur at the top of an object; curves pointing to the right occur at the right, etc. In many other cases, however, the preferred object-relative position is off angle, and when that is true the responses to circles will be low, and the responses to isolated arcs will be higher. (Responses to an entire shape with the arc at the preferred position would be higher still.) Thus, differential tuning for object-relative position explains most of the differences highlighted in Figures 4 and 5.

5) To support the 2-photon data, which surveys properties in a large number of cells, a comprehensive, more clearly structured data summary is required.

Apart from the issue raised above, the summary statements of the 2-Photon data are not particularly satisfying: the categories of orientation and curve-orientation are not very enlightening about underlying principles. Perhaps PCA analysis could be used to see the patterns of selectivity that are prevalent across the sample (perhaps after selecting the best orientation or collapsing across orientation). Another possibility would be, after rotating to a common best orientation, to superimpose all the stimuli with an opacity relating the firing rate. In the end this seems like a lost opportunity to follow up the influential Hegde and Van Essen, 2000, studies whose stimulus set parallels the one used here.

6) Incorporating any new results from the analyses above, the underlying major concepts need to be more consistently defined and applied across the paper.

Related to the technical issues raised above (points 3-5), the terminology in the manuscript is sometimes confusing. The authors propose that there is “microarchitecture” of selectivity for circles vs. curves. But then they appear to contradict themselves. For example (final paragraph of subsection “Microarchitectures of the curvature domains”) the authors talks of neurons having “preferred curve” stimuli; and in the same section, (Figure 4K) the authors state that these same neurons respond more strongly to circles than to their preferred curves. What does it mean to be selective for a “preferred curve” if the response to the “preferred” stimulus is consistently weaker than to a different stimulus?

7) Please clarify the definition of curvature domains and examine their periodicity

Another technical issue is the poor description of how curvature domains were defined and drawn. The "watershed" technique should be described in detail, and the domains should be drawn on the sections, especially the sections in Figure 5, where the domain boundaries determine the average response pattern.

Given the limited spatial sampling with 2-photon imaging, distance correlations metrics with the intrinsic optical imaging reflectance data (using the above requested single condition activity maps) would be nice to examine the periodicity of horizontal organization patterns. Given the stimulus array is that used in the 2-photon data, the absolute value of correlations is going to be different, but the key issue here is the spacing and size of "columns" and whether that depends on the type of stimulus you're looking at.

What bandpass filtering is used for intrinsic optical images (since that can "create" domains out of noise)? Can Fourier analysis be used to describe the size and spacing of domains?

8) There are also a number of places where the primary narrative of the paper seems to contradict itself. For example: starting with the Abstract, the authors make the important observation that “curvature” domains have “little overlap” with “orientation domains”. They repeat this point, e.g. in the Results; and again, in the legend of Figure 3H,: with clearly distinct “curvature” and “rectilinear” zones. On the other hand, the authors also make a point of saying that individual neurons have mixed responses – with individual neurons in “curvature” domains responding well to rectilinear contours (e.g. Figure 3P). The authors further amplify the issue of response diversity in Figure 5, with nearest-neighbor neurons showing strikingly different response. The authors should rewrite the Abstract and the relevant sections of the results to have a more consistent description of the degree of overlap and response diversity that they observe.

9) While these results are very interesting, they are of course restricted to a limited set (73?) of human-selected and human-curated stimuli. Nowhere does the analysis address the question as to whether these are close to the optimal stimuli for this region of cortex, even in the (unnatural) anesthetized state. The authors should have a more expanded discussion of these limitations, and how to extend the analysis further to get a sense of the “optimal” stimuli for this stage of cortical processing.

Furthermore, the stimulus set used parallels that used by Hedge and Van Essen, 2000, in V4, and subsequently in V1 and V2 as well. Given that we know that selectivity’s exist that cannot simply be explained by orientation filters, the paper could do a better job of emphasizing what is new and, in particular, the significance of finding functional organization. If, for example, the authors feel that the presence of functional organization for a particular parameter is especially important in understanding the computations that occur within in an area, or in the broader picture, the role that area plays in visual processing, they should state so, and why they believe this.

These issues should be discussed and laid out appropriately.

[Editors' note: further revisions were suggested prior to acceptance, as described below.]

Thank you very much for resubmitting your article "Curvature-processing domains in primate V4" for consideration by *eLife*. Your revised article has been reviewed by two peer reviewers, and the evaluation has been overseen by a Reviewing Editor and Joshua Gold as the Senior Editor. The following individual involved in review of your submission has agreed to reveal their identity: Aniruddha Das (Reviewer #2).

The reviewers have discussed the reviews with one another and the Reviewing Editor has drafted this decision to help you prepare a revised submission.

Summary:

Lu and colleagues present compelling new results on the architecture of curvature response strength and selectivity in macaque visual cortical area V4, in comparison to the lack thereof in earlier level visual areas V1 and V2. A particular strength is that the authors combine intrinsic-signal optical imaging (giving meso-scale resolution over a few mm of cortex) with 2-photon imaging (single cell resolution, over a few 100 microns; including 2-photon imaging at multiple cortical depths). The intrinsic signal data show distinct clustering of responsiveness to object fragments that constitute "intermediate" visual representations that are more complicated that the traditional single orientation columns in V1 or V2.

Revisions:

The authors have done an exemplary job in the revision, by addressing concerns raised and adding further analyses (clustering of single-cell-level response; PCA; more extended comparisons across classes of similar stimuli such as triangles / short oriented lines, as vs. circles / arcs; scaled to different sizes; filled / unfilled). Overall, these new analyses along with their original combination of intrinsic optical imaging and 2-photon imaging, showing the close spatial match between clusters identified separately by the two techniques make a compelling case for the specialised architecture the authors propose in the Discussion. There remain a small number of issues that would need to be addressed.

1) One disappointment is that some of the authors' responses didn't actually make into the manuscript or are "buried" in multi-panel supplementary figures. Specifically, the significance of relationship between surround suppression and curvature, and the PCA analysis (which proved interesting) are not particularly clear in the actual text of the manuscript. Also in light of some of the new analyses, the authors should probably cite some relevant literature about the existence of "sub-domain" specificity with domain-specific regions.

2) The direct correspondence between intrinsic and 2-photon imaging is certainly positive (correlation coefficients > 0), but remarkably poor (and less than one), especially given that, for many experiments, intrinsic maps are seen as veridical and generally have good correspondence with targeted electrophysiology recordings. We noticed in the response letter that all intrinsic maps were Gaussian blurred, if one does the same procedure to the 2-photon data, does the correlation improve? And, if not, what explains this result: perhaps, significant noise in one type of measurement, or more interestingly, a true lack of correspondence between haemodynamics and cellular level responses. If the later, that's a big deal given: it potentially suggests that all haemodynamic maps should be taken with "a grain of salt" with regards to reflecting neuronal response property organization. And of course, if this is true, and intrinsic imaging is that noisy, it potentially calls into question many of the intrinsic signal difference maps and results even within this manuscript. This needs to be explicitly addressed.

3) More discussion in the actual manuscript about surround suppression, and specifically the points that appear in the authors' response, would be good. This is especially important, because surround suppression has long been associated with V4 responses, and there have been claims that it is functionally organized. The authors' remarks in the response about the only things the difference maps having is common is curvature is probably the best that can be made with the current data set. But, on the other hand, given the strong differences in surround suppression and receptive field size shown in the supplementary figures, that previous reports of surround suppression and its organization in V4 might simply reflect curvature regions that were studied with inappropriate sub-optimal stimuli.

4) This is a really large data set of cellular responses, and the PCA analyses are really important, but again, it would be better for this to appear in the actual manuscript.

One reason in particular is the paper's discussion of "microorganization" or sub-domains within a curvature region. We found these discussions a little anecdotal, and we were wishing that a spatial autocorrelation analysis could be done with pairs of neurons as function of distance to reveal how big a "sub-domain" was, and in particular, how it compared with orientation columns, which of course are sub-domains within the rectiiinear domains. For example, what one could do with regard to PCA weights: PCA1 spatial autocorrelation. In a similar vein, there have been several reports of color-specific sub-domains with color-preference domains, and, especially given the classic association of V4 with color sensitivity, that literature seems relevant here in a discussion of how universal domain/sub-domain organization may be and its variation between different cortical areas.

5) Scaling of the intrinsic signal optical images. The authors' methods suggest that the SD used for the gray scale is calculated separately for each (SVM) map. Is that correct? If so, there is no way to compare the strengths of activation by different stimuli. It likely also explains why the single-condition maps (e.g. Figure 1—figure supplement 1, panels A,B,D,E,G,H) appear not only relatively featureless but also occupying a grayscale range similar to the backgrounds of the difference images. In absolute terms, the single-condition responses presumably are ~ an order of magnitude higher amplitude if they are dominated by stimulus-nonspecific haemodynamic responses? Showing the single-condition and difference images on the same absolute scale would be uninformative. But it would be useful to show all the difference images on a common scale that is not tied to the SD per image, to get a sense of the efficacies of different stimulus pairwise differences. This could be done as an additional set of panels in supplementary figures.

---

## [Author Response]

Essential revisions:1) To include single activation condition maps for the intrinsic optical imaging and explore whether map activations could be explained by simpler, underlying response patterns.The presented intrinsic optical imaging data comprise only difference maps, which could potentially obscure underlying results. For example, in Figure 1L, it could be that those 2 angles elicit very little localized responses, or that they elicit an identical pattern of activation, and from the difference map we don't know which is the case. It would be good to see some single condition activation maps.

We agree with the reviewer that difference maps show the differences between two conditions, not the true response patterns. In the revised manuscript, we have included single-condition maps (new Figure 1—figure supplement 1) for comparison.

As we can see, although some activation patterns can be observed from these single-condition maps, the overall contrast of these patterns was low, and the differences between these maps were small. There were almost no differences for stimuli presented at different orientations (new Figure 1—figure supplement 1 D and E, or G and H). However, calculated from these similar single-condition response patterns, the difference maps are not similar: strong patterns can be observed in some maps (C, L), and weak (F) or no (I) patterns in others. Thus, single-condition maps here are not reliable for revealing the differences of activation patterns for different stimuli. This might be due to the diffused feature-non-specific hemodynamic responses in V4, blood vessel noise, and the reliance on blank activation. In addition, single-condition maps were more vulnerable to experimental factors like the anesthesia level, the general activation level of the preparation etc., which may vary from case to case. Thus, we choose to rely on difference maps in this study.

That would also potentially help in investigating to what extent these responses can be explained by simple receptive field (RF) summation. For instance, there is a substantial literature in associating preferential responses to curves with end-stopped neurons: Hubel and Wiesel, in 1965 suggested that end-stopped, length tuned neurons could be useful for curvature detection and there was a nice explicit model of how that could happen by Dobbins et al. in 1987. But the manuscript makes little reference to any size tuning or surround suppression, and how that, in combination with classic orientation tuning, could create a cell that responds vigorously to curvature. This is even a greater concern given the well-established prevalence of size tuning, both electrophysiologically (1987) and with optical imaging (1997), in area V4. This has a huge impact on both novelty and interpretability; if you have a region of surround suppression (which we know exist) and it overlaps with an orientation region, this can look like a "curvature" region in that it will respond to a curve better than extended straight line or grating. But we would argue it's not a "curvature" region, since a short bar or grating without any curvature is actually the optimal stimulus. For example, can the response to triangles, on a pixel by pixel basis, be explained by simply adding or averaging the response the constituent orientations? Similarly, can the circle responses be explained by adding the responses to the half-circles?

The reviewer’s question is two-fold: First, at single-cell level, whether end-stopping properties can account for the neuron’s curvature responses we observed. We have shown that curvature neurons had smaller RFs than those of non-curvature neurons (Figure 3—figure supplement 1D and E). In this revision, we further analyzed and showed that curvature neurons had stronger surround suppression (Figure 3—figure supplement 1H and I). Thus, consistent with previous findings, curvature neurons did exhibit end-stopping properties. However, curvature neurons showed additional curve-specific preferences, including curve-orientation tuning, preference for curves over similarly-sized angle stimuli, etc. For circle-preferring neurons, they prefer circles (larger) than the half-circles (smaller). These additional features suggest that curvature neurons were not simple end-stopping neurons, although they exhibited end-stopping properties.

The same question was asked at the functional domain level, i.e., whether the curvature domains could be the overlap regions of orientation domains and surround-suppression regions in V4. In this study, curvature domains were revealed in difference maps, including: circles vs. triangles (Figure 1D), circles vs. bars (Figure 1G), curves vs. bars (Figure 1H), and curves vs. angles (Figure 1—figure supplement 2L). The two stimuli in each pair had similar sizes, and thus had similar surround suppression effects. Their contrast pattern in the difference maps could not be due to differences in suppression. The stimuli on the two sides also had balanced orientations, plus their maps had little overlap with the orientation maps (Figure 2D). The only consistent contrasts in these pairs of stimuli was the curvature feature. Thus, the curvature domains we found are unlikely the overlap regions of suppression and orientation domains.

2) To support the 2-photon results with convincing anatomical images.One technical issue is the lack of any good underlying anatomical images to support the 2-photon data. This is concerning because all the images provided here show cells that are filled in, but living cells should appear as rings representing the cell membrane. Filled in cells are dead or compromised, and this would be a critical caveat for all the response results reported here. All of the cells in Figure 3C-F appear to be filled in. This could be because (i) these images do not have sufficient resolution to show rings, (ii) the authors did some kind of smoothing of these images (smoothing is mentioned in Materials and methods), or (iii) the cells are filled with calcium, reflecting a dead or compromised state. This concern is further exacerbated by the strange cell shapes in Figures 4G and H and 5A, D, and G. The authors should show the underlying anatomical images for those panels and explain how they defined the odd shapes shown in them. In addition to showing anatomical images, the authors should quantify how many cells are filled or ring-like in their images, and clarify whether the cells for which they report tuning include filled cells (and how many).

Neurons in Figure 3C looked like filled was mainly due to the small size of the figure and the narrow clip range. In Author response image 1 we enlarged Figure 3C in panel A and also provided a larger clip-range version of the same image (panel B). It can be seen that most cell bodies appear ring-like.

GCaMP is expressed in the plasma and usually does not go into the nucleus (Tian et al., 2009; Chen et al., 2013). Thus, in ideal condition (the imaging focal plane cuts through the cell nucleus), the cells appear as rings. For neurons located slightly off the imaging focal plane, and their nucleus are not cut by the focal plane, the cells will also appear as smaller filled patches. In addition, Figure 3C was an average of all the frames over an experimental session, the preparation usually had some drift along the Z axis, the ring-like appearance will also be blurred due to this reason.

Fluorescent response images (e.g. Figure 3 D-E) were based on incremental fluorescence changes at the stimulus onset. Neurons in these images usually had responses at the center and appeared less ring-like. This fact, plus the smoothing procedure in cell identification (described in Materials and methods), made the final cell images filled shapes (e.g. Figure 5 G and H).

There were indeed some neurons having a high baseline fluorescence but weak visual responses. Since our cell identification was based on neurons’ onset responses, these neurons would not pass the significance test for further analysis.

Based on fluorescent responses, many identified neurons did not have a circular shape. This was mainly due to the shape of their fluorescence responses and/or the responses from attached fibers imaged in the same focal plane.

3) The reviewers would like to see more systematic and consistent analysis and representation across Figures 4. and 5 that allows the reader to better understand the fundamental, underlying tuning properties of neurons.A technical concern is that Figure 4 and Figure 5 appear to present the same phenomena, but with different analyses for different images. Figure 4G labels neurons according to whether they exhibited stronger responses to circles or arcs. Figure 5 distinguishes neurons by how closely they are correlated with the average response pattern for the local curvature domain. But the examples in Figures 5B,C,E,F,H,I make it clear that the differences between correlated and not correlated again have to do with relative responses to circles vs. arcs. All of these images in Figures 4 and 5 need to be analyzed in the same ways, with an emphasis on the more explanatory analyses related to tuning, rather than unexplained correlation.

We agree that the analysis in the original Figure 4 and 5 had some overlap on the same phenomena (i.e. different neurons responded differently to circles and arcs). In the revision, we replaced the original Figure 5 with a new figure (Figure 6) showing response matrices of neighboring neurons in 3 small regions (a circle subdomain, a curve-orientation subdomain, and a rectilinear region). These raw response patterns contain richest information, from which both commonality and diversity of the responses for the neighboring neurons can be observed.

In relations to the tuning properties underlying the impressive 2-photon data, the question arises to what extent can RF summation (for example, RF summation models that include non-linearities that have been applied to V4) explain responses across the stimulus array?

Due to the scope of this study, we have not evaluated the RF summation model in explaining the responses we observed. We agree, however, that this is an important question and needs to be explored. Based on our limited stimulus set, we observed complex response patterns for different neurons. Even for the neighboring neurons, they exhibited significant differences (Figure 6). Such a complexity seems difficult to be fully explained by a RF summation model. A recently study used a neural network approach and synthesized stimulus patterns that can maximally drove V4 neurons (Bashivan et al., 2019). These “optimal” stimulus patterns revealed complex RF structures of V4 neurons and suggest that additional complexity needs to be included in V4 RF modeling.

See also point 4 below, in relation to this issue of the underlying tuning properties of neurons.4) To reanalyze both datasets (intrinsic optical imaging and 2-Photon) in light of what is well-known about V4 tuning for fragments, not whole shapes, for curvature acuteness (which means responses to triangles reflect high curvature acuteness tuning) and object-relative position of contour fragments (which explains the differences between "circle" neurons and "curve" neurons, as well as the difference between triangle responses and angle responses, most of which we do not get to see). This should radically change the interpretation of Figures 1 and 2, and change the analysis emphasis of regions in the 2-photon section of the paper.The major technical and interpretational concern is that differences between response patterns throughout the paper are presented as though the neurons were tuned for circles in some cases, arcs in others, triangles in others, etc. This ignores well-established tuning dimensions in V4 that explain response patterns like these at a more basic level. Pasupathy and Connor, 1999; 2001 and 2002 and Carlson et al. (2011, Current Biology) clearly demonstrated that:i) V4 neurons respond to contour fragments, not complete shapes,ii) V4 neurons are strongly tuned for the object-relative position of those contour fragments,iii) V4 neurons are tuned for curvature acuteness, with a bias toward sharp curvatures,iv) V4 neurons are differentially tuned for convexity and concavity.By ignoring this literature, the authors might misinterpret many of their results:a) In Figures 1 and 2, responses to triangles are treated as representative of non-curvature regions. But, per point (iii) above, the points of those triangles will drive strong responses from the most acute curvature tuning regions. Thus, the curvature/triangle contrasts are more likely to represent organization for broad curvature vs. acute curvature (angles are geometrically just the limiting case of acute curvature) than a difference between curvature and straight lines. The same is true for line segments, whose terminations drive weaker but similar responses, and which show only weak differences from triangles and angles. The only valid contrast for finding curvature domains is curves vs. gratings, which reveals very different regions than triangles in Figure 1. This reliance on circle vs. triangle contrast has ramifications throughout the paper, because it is used to define curvature domains and thus bias the 2-photon studies away from regions of acute curvature tuning.

It is potentially possible that triangles we used in Figure 1 and 2 might represent acute curvatures and not for “rectilinear contour” representations. However, we found this was not the case after we examined many comparison maps. The responses to triangles were similar to those to short lines. For example, the triangle vs. straight line map was weak (Figure 1J), and the circle vs. triangle map and circle vs. straight line map were similar (Figure 1D, G). These indicate that responses elicited by triangles were mostly due to their line component, not the sharp angle component.

Following the reviewer’s suggestion, we compared “circle vs. gratings”, and “circle vs. triangle” maps (A and B in Author response image 2). In these two maps, the curvature domains were found at the similar locations. Thus, the V4 responses to the triangles were primarily driven by their straight-line components, not by their sharp angles. The weak contrast in “triangle vs. gratings” map (Author response image 2) also indicates the similarity of these two types of responses.

**Author response image 2. respfig2:** 

b) A similar problem appears in Figure 3, where the only response shown for the "rectilinear" region is a highly selective response for a triangle. Per point (ii) above, triangles provide a whole object context in which V4 neurons can exhibit their strong tuning for object-relative position of contour fragments, including high-curvature angles. This is why neuron 3 does not respond to any other rectilinear stimulus. If triangle responses like this are the basis for the green regions in Figure 3, then those regions are acute curvature regions, not non-curvature regions, and again the authors will be excluding high-curvature regions from their analyses in Figures 4 and 5.

We thank the reviewer for pointing this out. We re-examined cell 3 in the original Figure 3P and found that, in addition to its response to triangle, this cell did respond to line segments oriented in 45 degrees. The original figure did not include this stimulus due to the space limitation. Now we have expanded the figure and included this and other relevant stimuli (new Figure 3F). Thus, the triangle responses of this neuron were likely driven by the 45-degree line segment in the triangle, instead of its high-curvature angles. More examples neurons from the rectilinear regions were provided in a new figure (Figure 6C) in which similar responses patterns were observed.

Population-wise, neurons that responded better to bars than to curves were more likely located in the “rectilinear” regions in ISOI imaging (old Figure 3R, new Figure 4K).

Thus, consistent with the findings in ISOI results (see answers to reviewer’s question “a” above), the 2-photon results also indicate that, although triangle stimuli were used to map the rectilinear regions, neurons in these regions were mainly driven by the line segments of the rectilinear stimuli, not their sharp angles.

c) Per above, Figures 4 and 5 are restricted to clusters of broader curvature tuning, strongly biasing the analyses. This is undoubtedly one reason that many example neurons respond strongly to circles. The other major problem with these figures is that they are presented as evidence that many neurons in V4 are strongly selective for circles as whole stimuli, while others are strongly selective for arcs over circles. In fact, the references cited above make clear that neurons in V4 are invariably tuned for shape fragments (i), not whole shapes, and they are strongly tuned for fragment position within shapes (ii). This is why many neurons respond most strongly to circles, because they are tuned for an arc-shaped fragment, but respond much more strongly when that fragment is in the preferred object-relative position. This is frequently true for circles, because there is also a strong correlation between curvature orientation tuning and object position tuning, since convex curves pointing up usually occur at the top of an object; curves pointing to the right occur at the right, etc. In many other cases, however, the preferred object-relative position is off angle, and when that is true the responses to circles will be low, and the responses to isolated arcs will be higher. (Responses to an entire shape with the arc at the preferred position would be higher still.) Thus, differential tuning for object-relative position explains most of the differences highlighted in Figures 4 and 5.

The reviewer’s main concern here is that we separated curvature neurons into two sub-groups (circle-preferring and curve-orientation-preferring neurons), and why the circle responses were not explained in terms of position-related curvature responses.

We separated circle-preferring neurons from curve-orientation neurons based on two observations: First, these two sets of neurons formed separated spatial clusters on V4 surface (Figure 5 G and H) and differed in their curve orientation tunings. Second, as we can observe from the response matrices in the new Figure 6, that neurons in the circle subdomains (Figure 6A) responded best to circles, but very weakly to any of the half circles (the third, fourth columns from the right). Note that we used 8 orientations here instead of 4 in other shapes, thus likely covered all possible orientations. These neurons also appeared to have a gradual increase of responses to the completeness of the circle (from the right-most column to the middle). In addition, they all had modest responses to closed contour shapes (the four columns left to the circle). These closed shapes do not share any similar curve fragment, but have the closeness as a common feature. In particular, these neurons responded better to the offset-paired half circles (the second column on the right of the circle) than to the single half circles. Population results shown in Figure 5 K and L also confirm the above observation. These features were not easily explained by the selectivity to a curve fragment with a certain object-relative position.

Considering that these small patches of neurons were located in domains identified with ISOI and only occupied a small fraction of V4 surface, it is possible that these neurons were not well represented in the previous single-unit recordings.

5) To support the 2-photon data, which surveys properties in a large number of cells, a comprehensive, more clearly structured data summary is required.Apart from the issue raised above, the summary statements of the 2-Photon data are not particularly satisfying: the categories of orientation and curve-orientation are not very enlightening about underlying principles. Perhaps PCA analysis could be used to see the patterns of selectivity that are prevalent across the sample (perhaps after selecting the best orientation or collapsing across orientation). Another possibility would be, after rotating to a common best orientation, to superimpose all the stimuli with an opacity relating the firing rate. In the end this seems like a lost opportunity to follow up the influential Hegde and Van Essen, 2000, studies whose stimulus set parallels the one used here.

We thank the reviewer for this suggestion. We did PCA analysis on all the neurons’ responses (n=1556). The results were shown as new Figure 4—figure supplement 1. This data-driven analysis also reveals different response patterns for neurons inside and outside the curvature domains, which further confirmed our ISOI findings. We also performed PCA analysis on neurons inside curvature domains and found that circle neurons and curve-orientation neurons could also be separated by their PC coordinates (Figure 5—figure supplement 1).

6) Incorporating any new results from the analyses above, the underlying major concepts need to be more consistently defined and applied across the paper.Related to the technical issues raised above (points 3-5), the terminology in the manuscript is sometimes confusing. The authors propose that there is “microarchitecture” of selectivity for circles vs. curves. But then they appear to contradict themselves. For example (final paragraph of subsection “Microarchitectures of the curvature domains”) the authors talks of neurons having “preferred curve” stimuli; and in the same section, (Figure 4K) the authors state that these same neurons respond more strongly to circles than to their preferred curves. What does it mean to be selective for a “preferred curve” if the response to the “preferred” stimulus is consistently weaker than to a different stimulus?

We agree that the original categorization was not clear. In the revision, we have separated the “dual-preference neurons” from the “curve-orientation-preference neurons” so that these neurons do not have overlaps. We revised the figure (new Figure 5 G, I, J) and the relevant text accordingly.

7) Please clarify the definition of curvature domains and examine their periodicityAnother technical issue is the poor description of how curvature domains were defined and drawn. The "watershed" technique should be described in detail, and the domains should be drawn on the sections, especially the sections in Figure 5, where the domain boundaries determine the average response pattern.

In most part of this manuscript, curvature domains were obtained with a threshold (gray level lower than 0-2SD) on the circle vs. triangle map (e.g. Figure 1D). Domain coverage in Figure 2D and the neuron classification in Figure 4 and 5 (original figures) were all based on this method.

Only in estimating domain sizes (Figure 2E), we used watershed method in order to deal with the “connected domain” problem, in which two or more domains were connected and multiple peaks existed. With watershed method, each time we found the strongest domain, fill this domain with the threshold values and moved on to find the next strongest domain. This process was repeated until all domains were identified. We have added the details of watershed method in the Materials and methods section.

Given the limited spatial sampling with 2-photon imaging, distance correlations metrics with the intrinsic optical imaging reflectance data (using the above requested single condition activity maps) would be nice to examine the periodicity of horizontal organization patterns. Given the stimulus array is that used in the 2-photon data, the absolute value of correlations is going to be different, but the key issue here is the spacing and size of "columns" and whether that depends on the type of stimulus you're looking at.What bandpass filtering is used for intrinsic optical images (since that can "create" domains out of noise)? Can Fourier analysis be used to describe the size and spacing of domains?

We thank the reviewer’s suggestion. We have analyzed domain spacing and size with a domain-averaging method. In circle-triangle maps (e.g. Figure 1D), we located the peaks of each curvature domains and averaged all these domains with their peaks aligned. We did the same analysis on orientation domains based on the orientation maps (e.g. Figure 1E). The map in Author response image 3 shows the average maps for curvature domains (n=142) and orientation domains (n=218). We also plotted the normalized gray-level profiles of these two maps in panel C for comparison. The sizes of these two domains were very similar, which is consistent with the domain size results we obtained with a fitting method (Figure 2E). The length of the blue arrow is 0.518 mm, which is the measured curvature-domain size using a fitting method in the paper. It is approximately equal to the half-height width of the profile curves. Some details also can be observed from these profiles. For example, for orientation domain, there was a suppressive white ring around the black domain (diameter 1.17 mm), indicating there were opposite domains right next to these orientation domains (domains preferring orthogonal orientations). However, such feature was absent in the averaged curvature domain, suggesting that its opposite domains (rectilinear domains) were either not prominent or not adjacent.

**Author response image 3. respfig3:** 

Above results were obtained from difference maps. As we showed in Author response image 1, the signal-noise ratio in single-condition maps were too low to be analyzed with this method. We also tried Fourier analysis but the results were not as clear as those described above.

For the bandpass filter question the reviewer asked, we only used a small low-pass Gaussian filter (diameter: 5 pixels, SD: 1 pixel) to smooth the original ISOI maps. No high-pass filter was used. This procedure is unlikely to create domains out of noises.

8) There are also a number of places where the primary narrative of the paper seems to contradict itself. For example: starting with the Abstract, the authors make the important observation that “curvature” domains have “little overlap” with “orientation domains”. They repeat this point, e.g. in the Results; and again, in the legend of Figure 3H,: with clearly distinct “curvature” and “rectilinear” zones. On the other hand, the authors also make a point of saying that individual neurons have mixed responses – with individual neurons in “curvature” domains responding well to rectilinear contours (e.g. Figure 3P). The authors further amplify the issue of response diversity in Figure 5, with nearest-neighbor neurons showing strikingly different response. The authors should rewrite the Abstract and the relevant sections of the results to have a more consistent description of the degree of overlap and response diversity that they observe.

We took the reviewer’s advice and revised the narrative in Abstract and Results to make the meaning more consistent. We have also included new analysis (Figure 5 K and L, Figure 7) and examples of individual neuron’s response patterns (Figure 6) to better describe the seemly contradicting features.

9) While these results are very interesting, they are of course restricted to a limited set (73?) of human-selected and human-curated stimuli. Nowhere does the analysis address the question as to whether these are close to the optimal stimuli for this region of cortex, even in the (unnatural) anesthetized state. The authors should have a more expanded discussion of these limitations, and how to extend the analysis further to get a sense of the “optimal” stimuli for this stage of cortical processing.

We agree that the stimulus set we used is simple and limited, and could not address the “optimal stimulus” problem. In the revised Discussion (the second last paragraph), we described these limitations and our thoughts on how to improve estimation of “optimal stimuli”.

Furthermore, the stimulus set used parallels that used by Hedge and Van Essen, 2000, in V4, and subsequently in V1 and V2 as well. Given that we know that selectivity’s exist that cannot simply be explained by orientation filters, the paper could do a better job of emphasizing what is new and, in particular, the significance of finding functional organization. If, for example, the authors feel that the presence of functional organization for a particular parameter is especially important in understanding the computations that occur within in an area, or in the broader picture, the role that area plays in visual processing, they should state so, and why they believe this.

These issues should be discussed and laid out appropriately.

We thank the reviewer’s suggestions. In the revised Discussion (the last paragraph), we discussed more on the significance of the finding of functional organization and our reasons.

[Editors' note: further revisions were suggested prior to acceptance, as described below.]

Revisions:The authors have done an exemplary job in the revision, by addressing concerns raised and adding further analyses (clustering of single-cell-level response; PCA; more extended comparisons across classes of similar stimuli such as triangles / short oriented lines, as vs. circles / arcs; scaled to different sizes; filled / unfilled). Overall, these new analyses along with their original combination of intrinsic optical imaging and 2-photon imaging, showing the close spatial match between clusters identified separately by the two techniques make a compelling case for the specialised architecture the authors propose in the Discussion. There remain a small number of issues that would need to be addressed.1) One disappointment is that some of the authors' responses didn't actually make into the manuscript or are "buried" in multi-panel supplementary figures. Specifically, the significance of relationship between surround suppression and curvature, and the PCA analysis (which proved interesting) are not particularly clear in the actual text of the manuscript. Also in light of some of the new analyses, the authors should probably cite some relevant literature about the existence of "sub-domain" specificity with domain-specific regions.

We added surround suppression and PCA results into the main text (new Figure 4L and M, new Figure 5). We also discussed relevant findings about “sub-domains” in different domain-specific regions and cite the relevant papers (first paragraph in Discussion).

2) The direct correspondence between intrinsic and 2-photon imaging is certainly positive (correlation coefficients > 0), but remarkably poor (and less than one), especially given that, for many experiments, intrinsic maps are seen as veridical and generally have good correspondence with targeted electrophysiology recordings. We noticed in the response letter that all intrinsic maps were Gaussian blurred, if one does the same procedure to the 2-photon data, does the correlation improve? And, if not, what explains this result: perhaps, significant noise in one type of measurement, or more interestingly, a true lack of correspondence between haemodynamics and cellular level responses. If the later, that's a big deal given: it potentially suggests that all haemodynamic maps should be taken with "a grain of salt" with regards to reflecting neuronal response property organization. And of course, if this is true, and intrinsic imaging is that noisy, it potentially calls into question many of the intrinsic signal difference maps and results even within this manuscript. This needs to be explicitly addressed.

The correlation between ISOI and 2-photon results shows a general consistency, e.g. the overall similarity between two types of maps (maps in Figure 4A-I). The relatively low correlation values (r=0.42, 0.43) in Figure 4J and K may due to the nature of the signals in these two imaging methods. Specifically, 2-photon signals contained substantial diversity in neighboring neurons’ responses to the same contours (new Figure 7), while such diversity was not reflected in the hemodynamic signals. Due to optical blurring and the nature of the hemodynamical signal, ISOI maps were much blurred than 2-photon maps. The 5x5-Gaussian-bluring was to reduce the grainy noise before the alignment. In Figure 4J and K, a cell’s 2-photon response was averaged from all pixels that the cell body covered (> 30 pixels), thus, a further 5x5 filtering would not make much differences.

Thus, we think both technical factors (nature of the signals) and intrinsic diversity (cell-cell diversity) contribute to the relatively poor correlations between the two types of maps.

3) More discussion in the actual manuscript about surround suppression, and specifically the points that appear in the authors' response, would be good. This is especially important, because surround suppression has long been associated with V4 responses, and there have been claims that it is functionally organized. The authors' remarks in the response about the only things the difference maps having is common is curvature is probably the best that can be made with the current data set. But, on the other hand, given the strong differences in surround suppression and receptive field size shown in the supplementary figures, that previous reports of surround suppression and its organization in V4 might simply reflect curvature regions that were studied with inappropriate sub-optimal stimuli.

In the revised manuscript, we added surround suppression results into the main text and discussed how these results relevant to the previous findings (the eighth paragraph in Discussion).

Many studies have shown strong surround suppression in area V4. One of them also revealed a functional structure (S regions) associated with this feature (Ghose and Ts’o 1997). Although curvature domains have common properties as those S regions (stronger surround suppression), these two types of domains are unlikely the same. First, stimuli produce S regions did not contain any curviness, while curvature domains were only obtained with curvatures. Second, surround suppression is a more fundamental property than curvature preference. Although curvature neurons showed surround suppression, not all surround-suppressive neurons are curvature neurons. In area V4, surround suppression likely also involves, for example, texture, color, motion etc, in addition to contour shapes. Thus, we believe V4 surround suppression domains are independent domains that deserves further exploring.

4) This is a really large data set of cellular responses, and the PCA analyses are really important, but again, it would be better for this to appear in the actual manuscript.One reason in particular is the paper's discussion of "microorganization" or sub-domains within a curvature region. We found these discussions a little anecdotal, and we were wishing that a spatial autocorrelation analysis could be done with pairs of neurons as function of distance to reveal how big a "sub-domain" was, and in particular, how it compared with orientation columns, which of course are sub-domains within the rectiiinear domains. For example, what one could do with regard to PCA weights: PCA1 spatial autocorrelation. In a similar vein, there have been several reports of color-specific sub-domains with color-preference domains, and, especially given the classic association of V4 with color sensitivity, that literature seems relevant here in a discussion of how universal domain/sub-domain organization may be and its variation between different cortical areas.

We took the reviewers’ advice and analyzed the spatial periodicity of the sub-domains inside curvature domains. Specifically, we calculated the correlation of curve-orientation tuning of cell pairs inside the curvature domains, based on their responses to the 8 half-circle curves. Figure 6—figure supplement 2 plots how such correlation values vary with the cell-pair distance. It shows a trough around 360 μm and a second peak around 720 μm. Thus, the subdomain size is around 360 μm and the periodicity is around 720 μm. We had added this finding into the Results and discussed in Discussion. We did not use PCA results for such analysis since the curve-orientation was sorted before the PCA analysis.

5) Scaling of the intrinsic signal optical images. The authors' methods suggest that the SD used for the gray scale is calculated separately for each (SVM) map. Is that correct? If so, there is no way to compare the strengths of activation by different stimuli. It likely also explains why the single-condition maps (e.g. Figure 1—figure supplement 1, panels A,B,D,E,G,H) appear not only relatively featureless but also occupying a grayscale range similar to the backgrounds of the difference images. In absolute terms, the single-condition responses presumably are ~ an order of magnitude higher amplitude if they are dominated by stimulus-nonspecific haemodynamic responses? Showing the single-condition and difference images on the same absolute scale would be uninformative. But it would be useful to show all the difference images on a common scale that is not tied to the SD per image, to get a sense of the efficacies of different stimulus pairwise differences. This could be done as an additional set of panels in supplementary figures.

Different scaling methods (individual or common) were normally used for revealing different aspects of the map signals. Individual scaling is better at revealing spatial features, while common scaling can be used for signal strength comparisons.

In our manuscript, we used SVM maps, which was obtained through a non-linear process. The value of individual SVM pixel was proportional to its contribution to classification. Thus, it is not a simple representation of signal strength. Considering this, we used individual clipping to emphasize the spatial features of the domains. For analysis where comparing signal strengths were needed (e.g. Figure 2F), we used raw dR/R values instead of the SVM pixel values. We also tried a uniform clipping for Figure 1, the general appearance was similar to the individually clipped one in the manuscript.